# GENERATIVE AUGMENTED FLOW NETWORKS

**Ling Pan**[1,2], **Dinghuai Zhang**[1,2], **Aaron Courville**[1,2,4], **Longbo Huang**[3], **Yoshua Bengio**[1,2,4]
[1]Mila - Québec AI Institute [2]Université de Montréal [3]Tsinghua University
[4]CIFAR AI Chair
{penny.ling.pan}@gmail.com

## ABSTRACT

The Generative Flow Network (Bengio et al., 2021b, GFlowNet) is a probabilistic framework where an agent learns a stochastic policy for object generation, such that the probability of generating an object is proportional to a given reward function. Its effectiveness has been shown in discovering high-quality and diverse solutions, compared to reward-maximizing reinforcement learning-based methods. Nonetheless, GFlowNets only learn from rewards of the terminal states, which can limit its applicability. Indeed, intermediate rewards play a critical role in learning, for example from intrinsic motivation to provide intermediate feedback even in particularly challenging sparse reward tasks. Inspired by this, we propose Generative Augmented Flow Networks (GAFlowNets), a novel learning framework to incorporate intermediate rewards into GFlowNets. We specify intermediate rewards by intrinsic motivation to tackle the exploration problem in sparse reward environments. GAFlowNets can leverage edge-based and state-based intrinsic rewards in a joint way to improve exploration. Based on extensive experiments on the GridWorld task, we demonstrate the effectiveness and efficiency of GAFlowNet in terms of convergence, performance, and diversity of solutions. We further show that GAFlowNet is scalable to a more complex and large-scale molecule generation domain, where it achieves consistent and significant performance improvement.

## 1 INTRODUCTION

Deep reinforcement learning (RL) has achieved significant progress in recent years with particular success in games (Mnih et al., 2015; Silver et al., 2016; Vinyals et al., 2019). RL methods applied to the setting where a reward is only given at the end (*i.e.*, terminal states) typically aim at maximizing that reward function for learning the optimal policy. However, diversity of the generated states is desirable in a wide range of practical scenarios including molecule generation (Bengio et al., 2021a), biological sequence design (Jain et al., 2022b), recommender systems (Kunaver & Požrl, 2017), dialogue systems (Zhang et al., 2020), etc. For example, in molecule generation, the reward function used in in-silico simulations can be uncertain and imperfect itself (compared to the more expensive in-vivo experiments). Therefore, it is not sufficient to only search the solution that maximizes the return. Instead, it is desired that we sample many high-reward candidates, which can be achieved by sampling them proportionally to the reward of each terminal state.

Interestingly, GFlowNets (Bengio et al., 2021a;b) learn a stochastic policy to sample composite objects $\mathbf{x} \in \mathcal{X}$ with probability proportional to the return $R(\mathbf{x})$. The learning paradigm of GFlowNets is different from other RL methods, as it is explicitly aiming at modeling the diversity in the target distribution, *i.e.*, all the modes of the reward function. This makes it natural for practical applications where the model should discover objects that are both interesting and diverse, which is a focus of previous GFlowNet works (Bengio et al., 2021a;b; Malkin et al., 2022; Jain et al., 2022b).

Yet, GFlowNets only learn from the reward of the terminal state, and do not consider intermediate rewards, which can limit its applicability, especially in more general RL settings. Rewards play a critical role in learning (Silver et al., 2021). The tremendous success of RL largely depends on the reward signals that provide intermediate feedback. Even in environments with sparse rewards, RL agents can motivate themselves for efficient exploration by intrinsic motivation, which augments the sparse extrinsic learning signal with a dense intrinsic reward at each step. Our focus in this paper is

precisely on introducing such intermediate intrinsic rewards in GFlowNets, since they can be applied even in settings where the extrinsic reward is sparse (say non-zero only on a few terminal states).

Inspired by this missing element of GFlowNets, we propose a new GFlowNet learning framework that takes intermediate feedback signals into account to provide an exploration incentive during training. The notion of flow in GFlowNets (Bengio et al., 2021a;b) refers to a marginalized quantity that sums rewards over all downstream terminal states following a given state, while sharing that reward with other states leading to the same terminal states. Apart from the existing flows in the network, we introduce augmented flows as intermediate rewards. Our new framework is well-suited for sparse reward tasks by considering intrinsic motivation as intermediate rewards, where the training of GFlowNet can get trapped in a few modes, since it may be difficult for it to discover new modes based on those it visited (Bengio et al., 2021b).

We first propose an edge-based augmented flow, based on the incorporation of an intrinsic reward at each transition. However, we find that although it improves learning efficiency, it only performs local exploration and still lacks sufficient exploration ability to drive the agent to visit solutions with zero rewards. On the other hand, we find that incorporating intermediate rewards in a state-based manner (Bengio et al., 2021b) can result in slower convergence and large bias empirically, although it can explore more broadly. Therefore, we propose a joint way to take both edge-based and state-based augmented flows into account. Our method can improve the diversity of solutions and learning efficiency by reaping the best from both worlds. Extensive experiments on the GridWorld and molecule domains that are already used to benchmark GFlowNets corroborate the effectiveness of our proposed framework. The code is publicly available at `https://github.com/ling-pan/GAFN`.

The main contributions of this paper are summarized as follows:

- We propose a novel GFlowNet learning framework, dubbed Generative Augmented Flow Networks (GAFlowNet), to incorporate intermediate rewards, which are represented by augmented flows in the flow network.

- We specify intermediate rewards by intrinsic motivation to deal with the exploration of state space for GFlowNets in sparse reward tasks. We theoretically prove that our augmented objective asymptotically yields an unbiased solution to the original formulation.

- We conduct extensive experiments on the GridWorld domain, demonstrating the effectiveness of our method in terms of convergence, diversity, and performance. Our method is also general, being applicable to different types of GFlowNets. We further extend our method to the larger-scale and more challenging molecule generation task, where our method achieves consistent and substantial improvements over strong baselines.

## 2 BACKGROUND

Consider a directed acyclic graph (DAG) $G = (\mathcal{S}, \mathbb{A})$, where $\mathcal{S}$ denotes the state space, and $\mathbb{A}$ represents the action space, which is a subset of $\mathcal{S} \times \mathcal{S}$. We denote the vertex $\mathbf{s}_0 \in \mathcal{S}$ to be the initial state with no incoming edges, while the vertex $\mathbf{s}_f$ without outgoing edges is called the sink state, and state-action pairs correspond to edges. The goal for GFlowNets is to learn a stochastic policy $\pi$ that can construct discrete objects $\mathbf{x} \in \mathcal{X}$ with probability proportional to the reward function $R : \mathcal{X} \to \mathbb{R}_{\geq 0}$, i.e., $\pi(\mathbf{x}) \propto R(\mathbf{x})$. GFlowNets construct objects sequentially, where each step adds an element to the construction. We call the resulting sequence of state transitions from the initial state to a terminal state $\tau = (\mathbf{s}_0 \to \cdots \to \mathbf{s}_n)$ a trajectory, where $\tau \in \mathcal{T}$ with $\mathcal{T}$ denoting the set of trajectories. Bengio et al. (2021a) define a trajectory flow $F : \mathcal{T} \to \mathbb{R}_{\geq 0}$. Let $F(\mathbf{s}) = \sum_{\tau \ni \mathbf{s}} F(\tau)$ define a state flow for any state $\mathbf{s}$, and $F(\mathbf{s} \to \mathbf{s}') = \sum_{\tau \ni \mathbf{s} \to \mathbf{s}'} F(\tau)$ defines the edge flow for any edge $\mathbf{s} \to \mathbf{s}'$. The trajectory flow induces a probability measure $P(\tau) = \frac{F(\tau)}{Z}$, where $Z = \sum_{\tau \in \mathcal{T}} F(\tau)$ denotes the total flow. We then define the corresponding forward policy $P_F(\mathbf{s}'|\mathbf{s}) = \frac{F(\mathbf{s} \to \mathbf{s}')}{F(\mathbf{s})}$ and the backward policy $P_B(\mathbf{s}|\mathbf{s}') = \frac{F(\mathbf{s} \to \mathbf{s}')}{F(\mathbf{s}')}$. The flows can be considered as the amount of water flowing through edges (like pipes) or states (like tees connecting pipes) (Malkin et al., 2022), with $R(\mathbf{x})$ the amount of water through terminal state $\mathbf{x}$, and $P_F(\mathbf{s}'|\mathbf{s})$ the relative amount of water flowing in edges outgoing from $\mathbf{s}$.

## 2.1 GFLOWNETS TRAINING CRITERION

We call a flow consistent if it satisfies the flow matching constraint for all internal states $\mathbf{s}$, *i.e.*, $\sum_{\mathbf{s}'' \to \mathbf{s}} F(\mathbf{s}'' \to \mathbf{s}) = F(\mathbf{s}) = \sum_{\mathbf{s} \to \mathbf{s}'} F(\mathbf{s} \to \mathbf{s}')$, which means that the incoming flows equal the outgoing flows. Bengio et al. (2021a) prove that for a consistent flow $F$ where the terminal flow is set to be the reward, the forward policy can sample objects $x$ with probability proportional to $R(\mathbf{x})$.

**Flow matching (FM).** Bengio et al. (2021a) propose to approximate the edge flow by a model $F_\theta(\mathbf{s}, \mathbf{s}')$ parameterized by $\theta$ following the FM objective, *i.e.*, $\mathcal{L}_{\mathrm{FM}}(\mathbf{s}) = (\log \sum_{(\mathbf{s}'' \to \mathbf{s}) \in \mathcal{A}} F_\theta(\mathbf{s}'', \mathbf{s}) - \log \sum_{(\mathbf{s} \to \mathbf{s}') \in \mathcal{A}} F_\theta(\mathbf{s}, \mathbf{s}'))^2$ for non-terminal states. At terminal states, a similar objective encourages the incoming flow to match the corresponding reward. The objective is optimized using trajectories sampled from a training policy $\pi$ with full support such as a tempered version of $P_{F_\theta}$ or a mixture of $P_{F_\theta}$ with a uniform policy $U$, *i.e.*, $\pi_\theta = (1 - \epsilon)P_{F_\theta} + \epsilon \cdot U$, This is similar to $\epsilon$-greedy and entropy-regularized strategies in RL to improve exploration. Bengio et al. (2021a) prove that if we reach a global minimum of the expected loss function and the training policy $\pi_\theta$ has full support, then GFlowNet samples from the target distribution.

**Detailed balance (DB).** Bengio et al. (2021b) propose the DB objective to avoid the computationally expensive summing operation over the parents or children of states. For learning based on DB, we train a neural network with a state flow model $F_\theta$, a forward policy model $P_{F_\theta}(\cdot|\mathbf{s})$, and a backward policy model $P_{B_\theta}(\cdot|\mathbf{s})$ parameterized by $\theta$. The optimization objective is to minimize $\mathcal{L}_{\mathrm{DB}}(\mathbf{s}, \mathbf{s}') = (\log(F_\theta(\mathbf{s})P_{F_\theta}(\mathbf{s}'|\mathbf{s})) - \log(F_\theta(\mathbf{s}')P_{B_\theta}(\mathbf{s}|\mathbf{s}')))^2$. It also samples from the target distribution if a global minimum of the expected loss is reached and $\pi_\theta$ has full support.

**Trajectory balance (TB).** Malkin et al. (2022) propose the TB objective for faster credit assignment and learning over longer trajectories. The loss function for TB is $\mathcal{L}_{\mathrm{TB}}(\tau) = (\log(Z_\theta \prod_{t=0}^{n-1} P_{F_\theta}(\mathbf{s}_{t+1}|\mathbf{s}_t)) - \log(R(\mathbf{x}) \prod_{t=0}^{n-1} P_{B_\theta}(\mathbf{s}_t|\mathbf{s}_{t+1})))^2$, where $Z_\theta$ is a learnable scalar.

# 3 RELATED WORK

**GFlowNets.** Since the proposal of GFlowNets (Bengio et al., 2021a), there has been an increasing interest in improving (Bengio et al., 2021b; Malkin et al., 2022), understanding, and applying this framework to practical scenarios. It is a general-purpose high-level probabilistic inference framework, and induces fruitful applications (Zhang et al., 2022a;c; Deleu et al., 2022; Jain et al., 2022a). Pan et al. (2023a) propose Forward-Looking GFlowNets by incorporating local credits, and makes it possible to be trained with incomplete trajectories (which is a requirement for previous GFlowNet methods). Pan et al. (2023b) and Zhang et al. (2023) recently generalize GFlowNets to the more general stochastic environments with stochastic transition dynamics or rewards. However, previous works only consider learning based on the terminal reward, which can make it difficult to provide a good training signal for intermediate states, especially when the reward is sparse (*i.e.*, significantly non-zero in only a tiny fraction of the terminal states).

**Reinforcement learning (RL).** Different from GFlowNets that aim to sample proportionally to the reward function, RL learns a reward-maximization policy. Entropy-regularized RL (Attias, 2003; Ziebart, 2010; Haarnoja et al., 2017; 2018; Zhang et al., 2022b) introduces entropy regularization in the learning objectives, which learns with the Boltzmann softmax policy and the log-sum-exp operator in value function updates (Schulman et al., 2017a; Asadi & Littman, 2017; Pan et al., 2020). Although it can improve diversity, this is limited to tree-structured DAGs. This is because it could only sample a terminal state $\mathbf{x}$ in proportion to the sum of rewards over all trajectories leading to $\mathbf{x}$. It can fail on general (non-tree) DAGs (Bengio et al., 2021a) for which the same terminal state $\mathbf{x}$ can be obtained with a potentially large number of trajectories (and a very different number of trajectories for different $\mathbf{x}$'s).

**Intrinsic motivation.** There has been a line of research to incorporate intrinsic motivation (Pathak et al., 2017; Burda et al., 2018; Zhang et al., 2021) for improving exploration in RL. Yet, such ideas have not been explored with GFlowNets because the current mathematical framework of GFlowNets only allows for terminal rewards, unlike the standard RL frameworks. This deficiency as well as the potential of introducing intrinsic intermediate rewards motivates this paper.

# 4  GENERATIVE AUGMENTED FLOW NETWORKS

The potential difficulty in learning only from the terminal reward is related to the challenge of sparse rewards in RL, where most states do not provide an informative reward. We demonstrate the sparse reward problem for GFlowNets and reveal interesting findings based on the GridWorld task (as shown in Figure 4) with sparse rewards. Specifically, the agent only receives a reward of $+1$ only when it reaches one of the 3 goals located around the corners of the world (except the starting state corner) with size $H \times H$ (with $H \in \{64, 128\}$), and the reward is 0 otherwise. A more detailed description of the task can be found in Section 5.1. We evaluate the number of modes discovered by the GFlowNet trained with TB, following Bengio et al. (2021a). As summarized in Figure 1, GFlowNet training can get trapped in a subset of the modes. Therefore, it remains a critical challenge for GFlowNets to efficiently learn when the reward signal is sparse and non-informative.

On the other hand, there has been recent progress with intrinsic motivation methods (Pathak et al., 2017; Burda et al., 2018) to improve exploration of RL algorithms, where the agent learns from both a sparse extrinsic reward and a dense intrinsic bonus at each step. Building on this, we aim to address the exploration challenge of GFlowNets by enabling intermediate rewards in GFlowNets and thus intrinsic rewards.

We now propose our learning framework, which is dubbed Generative Augmented Flow Network (GAFlowNet), to take intermediate rewards into consideration.

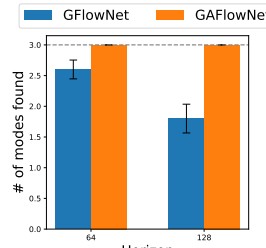

Figure 1: Comparison of GFlowNet and our augmented (GAFlowNet) method in Gridworld with sparse rewards .

## 4.1  EDGE-BASED INTERMEDIATE REWARD AUGMENTATION

We start our derivation from the flow matching consistency constraint, to take advantage of the insights brought by the water flow metaphor as discussed in Section 2. By incorporating intermediate rewards $r(\mathbf{s}_t \to \mathbf{s}_{t+1})$ for transitions from states $\mathbf{s}_t$ to $\mathbf{s}_{t+1}$ into the flow matching constraint, we obtain

$$\sum_{\mathbf{s}_{t-1}} F(\mathbf{s}_{t-1} \to \mathbf{s}_t) = F(\mathbf{s}_t) = \sum_{\mathbf{s}_{t+1}} \left[ F(\mathbf{s}_t \to \mathbf{s}_{t+1}) + r(\mathbf{s}_t \to \mathbf{s}_{t+1}) \right] \tag{1}$$

by considering an extra flow $r(\mathbf{s}_t \to \mathbf{s}_{t+1})$ going out of the transition $\mathbf{s}_t \to \mathbf{s}_{t+1}$. Based on Eq. (1), we define the corresponding forward and backward policies

$$P_F(\mathbf{s}_t | \mathbf{s}_{t-1}) = \frac{F(\mathbf{s}_{t-1} \to \mathbf{s}_t) + r(\mathbf{s}_{t-1} \to \mathbf{s}_t)}{F(\mathbf{s}_{t-1})}, \quad P_B(\mathbf{s}_{t-1} | \mathbf{s}_t) = \frac{F(\mathbf{s}_{t-1} \to \mathbf{s}_t)}{F(\mathbf{s}_t)}. \tag{2}$$

Combining these, we obtain the detailed balance objective with the incorporation of intermediate rewards as

$$F(\mathbf{s}_{t-1}) P_F(\mathbf{s}_t | \mathbf{s}_{t-1}) = P_B(\mathbf{s}_{t-1} | \mathbf{s}_t) F(\mathbf{s}_t) + r(\mathbf{s}_{t-1} \to \mathbf{s}_t). \tag{3}$$

Finally, we have our resulting edge-based reward augmented learning objective for trajectory balance as in Eq. (4) via a telescoping calculation upon Eq. (3), where $\mathbf{x} = \mathbf{s}_n$, and $Z = \sum_{\mathbf{s}_{t-1} \to \mathbf{s}_t} r(\mathbf{s}_{t-1} \to \mathbf{s}_t) + \sum_{\mathbf{x}} R(\mathbf{x})$ is the augmented total flow. Detailed derivation can be found in Appendix A.

$$Z \prod_{t=0}^{n-1} P_F(\mathbf{s}_{t+1} | \mathbf{s}_t) = R(\mathbf{x}) \prod_{t=0}^{n-1} \left[ P_B(\mathbf{s}_t | \mathbf{s}_{t+1}) + \frac{r(\mathbf{s}_t \to \mathbf{s}_{t+1})}{F(\mathbf{s}_{t+1})} \right]. \tag{4}$$

We explain the semantics of Eq. (4) in Figure 2(a). For the transition from an internal state (yellow circles) $\mathbf{s}_t$ to the $i$-th next state $\mathbf{s}_{t+1}^i$, we associate $\mathbf{s}_{t+1}^i$ with a special state $\hat{\mathbf{s}}_{t+1}^i$ (red circle) with pseudo-exit. Specifically, from the state $\mathbf{s}_t$, we choose associated next states $\hat{\mathbf{s}}_{t+1}$ with probability $(F(\mathbf{s}_t \to \mathbf{s}_{t+1}) + r(\mathbf{s}_t \to \mathbf{s}_{t+1})) / F(\mathbf{s}_t)$ according to the forward policy in Eq. (2). At the associated next state $\hat{\mathbf{s}}_{t+1}$, we "virtually" choose the sink state (purple circles) $\mathbf{s}_f$ with probability $r(\mathbf{s}_t \to \mathbf{s}_{t+1}) / F(\mathbf{s}_t)$, or we choose the next state $\mathbf{s}_{t+1}$ with probability $F(\mathbf{s}_t \to \mathbf{s}_{t+1}) / F(\mathbf{s}_t)$. Adding them together and multiplying these probabilities by the incoming flow $F(\mathbf{s}_t)$, we have $F(\mathbf{s}_t \to \mathbf{s}_{t+1}) + r(\mathbf{s}_t \to \mathbf{s}_{t+1})$. Therefore, considering all possible next states, we have the augmented flow consistency equation (incoming flow = outgoing flow) as in Eq. (1). The intermediate rewards $r(\mathbf{s}_t \to \mathbf{s}_{t+1})$ are similar to transitions into a pseudo-exit that is never taken but still attracts

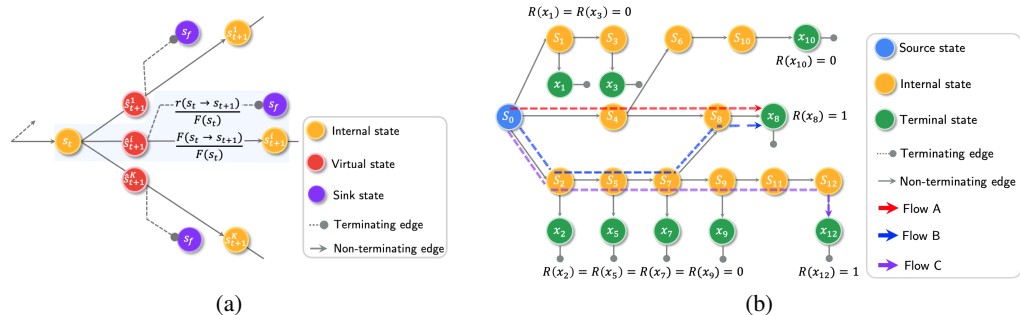

(a)                    (b)

Figure 2: (a) Edge-based reward augmentation can be seen as introducing an augmented flow of amount $r(\mathbf{s}_t \to \mathbf{s}_{t+1})$ towards a pseudo-exit to the sink state (that we never actually take) at every transition step. (b) For tasks with sparse rewards, agents can easily get stuck at a few modes (*e.g.*, $\mathbf{x}_8$). Our proposed method motivates the agent to discover unexplored states and trajectories to find diverse sets of modes (*i.e.*, $\mathbf{x}_{12}$) by increasing the probability of visiting alternative transitions in proportion to all the transitions reachable from there. Note that we omit the sink state from terminating edges for simplicity.

larger probabilities into its ancestors in the DAG. From the water analogy, it can be considered that the flow from $\mathbf{s}_t$ to $\mathbf{s}_{t+1}$ (and thus the probability of choosing that transition) is augmented by all the intermediate rewards due to pseudo-exits in all the accessible downstream transitions.

In contrast to simply adding a constant uniform probability to every action (which is commonly used for exploration with GFlowNets), the pseudo-exit intermediate rewards have an effect that is not local. In addition, we can specify non-uniform intermediate rewards as intrinsic motivation $r(\mathbf{s}_t \to \mathbf{s}_{t+1})$ with novelty-based methods (Pathak et al., 2017; Burda et al., 2018) to better tackle exploration in sparse reward tasks. Although how to accurately measure the novelty degree remains an open problem, it has been shown that random network distillation (RND) (Burda et al., 2018) is a simple yet effective method for encouraging the agent to visit states of interest. We use the scaled difference between the predicted features by a trainable state encoder and a random fixed state encoder as the intrinsic rewards based on RND, *i.e.*, $\alpha \|\phi(\mathbf{s}) - \bar{\phi}_{\text{random}}(\mathbf{s})\|_2$, where $\alpha$ is a reward scaling parameter that controls the degree of exploration. The random distillation network is trained by minimizing such differences. Therefore, the novelty measure is generally smaller for more often seen states or similar states. The overall training procedure is shown in Algorithm 1 by substituting the augmented trajectory balance loss according to Eq. (4).

We now demonstrate the conceptual advantage of edge-based reward augmentation which specifies intermediate rewards by intrinsic motivation in a sparse reward task for exploration in Figure 2(b). It depicts a flow network Markov decision process (MDP) with sparse rewards, where only $R(\mathbf{x}_8)$ and $R(\mathbf{x}_{12})$ are 1 and other terminal rewards are all 0. Consider the case where the agent had already discovered solution $\mathbf{x}_8$ with the red flow A. Since the rewards of most other solutions are 0, it can easily get trapped in the mode of $\mathbf{x}_8$, and thus fails to discover other solutions. Nonetheless, our edge-based reward augmentation could motivate the agent to discover other paths (*e.g.*, the blue flow B) to $\mathbf{x}_8$, which can be beneficial for the agent to discover other solutions (*e.g.*, $\mathbf{x}_{12}$) subsequently.

Following the evaluation scheme in (Bengio et al., 2021a), we summarize the number of discovered modes and the empirical $L_1$ error for GFlowNet and GAFlowNet with edge-based reward augmentation in Figure 3. The figure also includes the state-based and joint objectives introduced in later sections (Sections 4.2 and 4.3) for completeness. The $L_1$ error is defined as $\mathbb{E}\left[\|p(\mathbf{x}) - \pi(\mathbf{x})\|\right]$, where $p(\mathbf{x}) = R(\mathbf{x})/Z$ denotes the true reward distribution, and we estimate $\pi$ by repeated sampling and summarizing frequencies for visitation of each

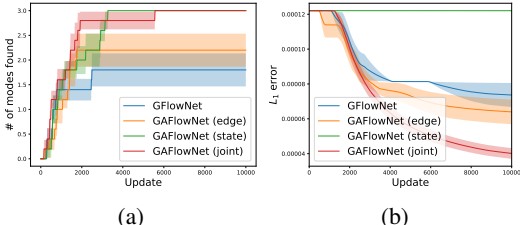

(a)                    (b)

Figure 3: Different reward augmentations proposed in Sections 4.1-4.3. (a) Diversity metric: the number of modes found. (b) Distribution fitness metric: empirical $L_1$ error.

possible state $\mathbf{x}$. As shown, GAFlowNet (edge) is able to discover more modes as opposed to a standard GFlowNet and improves diversity. In addition, it learns more efficiently and leads to a smaller level of $L_1$ error.

## 4.2 STATE-BASED INTERMEDIATE REWARD AUGMENTATION

Although the learning framework of edge-based reward augmentation is able to improve the diversity of solutions found, it still fails to discover all of the modes as shown in Figure 3. We hypothesize that this is due to its "local" exploration ability, where it is able to consider different paths to solution $\mathbf{x}_i$ with non-zero rewards. However, it fails to sufficiently motivate the agent to globally explore solutions whose rewards may be zero. Therefore, it can still get trapped in a few modes, lacking sufficient exploration ability to discover other modes.

Different from the edge-based reward augmentation, Bengio et al. (2021b) defines a *trajectory return* as the sum of intermediate rewards in a state-based reward augmentation manner. Specifically, state-based reward augmentation for trajectory balance yields the following criterion

$$Z \prod_{t=0}^{n-1} P_F(\mathbf{s}_{t+1}|\mathbf{s}_t) = \left[R(\mathbf{x}) + \sum_{t=0}^{n-1} r(\mathbf{s}_t \to \mathbf{s}_{t+1})\right] \prod_{t=0}^{n-1} P_B(\mathbf{s}_t|\mathbf{s}_{t+1}), \tag{5}$$

and we also use RND for intrinsic rewards. Such an objective directly motivates the agent to explore different terminate states in a more global way (*e.g.*, $\mathbf{x}_2$, $\mathbf{x}_5$, $\mathbf{x}_7$, $\mathbf{x}_9$ in Figure 2(b), which are beneficial for discovering $\mathbf{x}_{12}$). As shown in Figure 3(a), it is able to discover all the modes, exhibiting great diversity. However, it explicitly changes the underlying target probability distribution, and is directly and highly affected by the length of the trajectory. Therefore, this leads to much slower convergence as demonstrated in Figure 3(b).

## 4.3 JOINT INTERMEDIATE REWARD AUGMENTATION

As discussed above, the state-based reward augmentation is effective in improving diversity but fails to fit the target distribution efficiently. On the other hand, edge-based reward augmentation performs more efficiently, but lacks sufficient exploration ability which cannot discover all the modes.

Therefore, we propose a joint method to take both state and edge-based intermediate reward augmentation into account to reap the best from both worlds. Specifically, we redefine the trajectory return as the sum of the terminal reward and the intrinsic reward for the terminal state only. This can be considered as we augment the extrinsic terminal reward with its curiosity degree. On the other hand, we include intrinsic rewards for internal states according to the edge-based reward augmentation. This integration inherits the merits of both state and edge-based reward augmentation, which makes it possible to improve exploration in a more global way while learning more efficiently.

$$Z \prod_{t=0}^{n-1} P_F(\mathbf{s}_{t+1}|\mathbf{s}_t) = [R(\mathbf{x}) + r(\mathbf{s}_n)] \prod_{t=0}^{n-1} \left[P_B(\mathbf{s}_t|\mathbf{s}_{t+1}) + \frac{r(\mathbf{s}_t \to \mathbf{s}_{t+1})}{F(\mathbf{s}_{t+1})}\right]. \tag{6}$$

Our resulting flow consistency constraint is shown in Eq. (6) where $Z$ is the augmented total flow $\sum_{\mathbf{x}} R(\mathbf{x}) + \sum_{\mathbf{s}_{t-1} \to \mathbf{s}_t} r(\mathbf{s}_{t-1} \to \mathbf{s}_t)$. Our new optimization objective $\mathcal{L}_{\text{GAFlowNet}}(\tau)$ is Eq. (7) which is trained by Algorithm 1.

$$\left(\log \left(Z \prod_{t=0}^{n-1} P_F(\mathbf{s}_{t+1}|\mathbf{s}_t)\right) - \log \left([R(\mathbf{x}) + r(\mathbf{s}_n)] \prod_{t=0}^{n-1} \left[P_B(\mathbf{s}_t|\mathbf{s}_{t+1}) + \frac{r(\mathbf{s}_t \to \mathbf{s}_{t+1})}{F(\mathbf{s}_{t+1})}\right]\right)\right)^2 \tag{7}$$

In Theorem 1, we theoretically justify that the resulting joint augmentation method leads to an unbiased solution to the original formulation asymptotically. The proof can be found in Appendix B. Note that we employ RND (Burda et al., 2018) for the intrinsic rewards, which decrease as the agent has more knowledge about the state.

**Theorem 1.** *Suppose that $\forall \tau, \mathcal{L}_{\text{GAFlowNet}}(\tau) = 0$, and $\forall \mathbf{x}, R(\mathbf{x}) + r(\mathbf{x}) > 0$. When edge-based intrinsic rewards converge to 0, we have that (1) $P(\mathbf{x}) = \frac{R(\mathbf{x}) + r(\mathbf{x})}{\sum_{\mathbf{x}} [R(\mathbf{x}) + r(\mathbf{x})]}$; (2) If state-based intrinsic rewards converge to 0, then $P(\mathbf{x})$ is an unbiased sample distribution.*

---

**Algorithm 1** Generative Augmented Flow Networks.

---

1: Initialize the forward and backward policies $P_F$, $P_B$, learnable parameter $Z$, and state flow $F$
2: Initialize the random fixed target network $\bar{\phi}$ and the predictor network $\phi$
3: **for** each training step $t = 1$ to $T$ **do**
4:     Collect a batch of $B$ trajectories $\tau = \{\mathbf{s}_0 \rightarrow \cdots \rightarrow \mathbf{s}_n\}$ based on the forward policy $P_F$
5:     Compute intrinsic rewards $r$ for each sample in the batch of trajectories based on the random target network $\bar{\phi}$ and the predictor network $\phi$
6:     Update the GAFlowNet model according to the augmented trajectory balance loss in Eq. (7)
7:     Update the predictor network $\phi$ by minimizing $||\bar{\phi}(\mathbf{s}) - \phi(\mathbf{s})||_2$

---

As shown in Figure 3, the joint method is able to discover all of the modes. In addition, it converges to the smallest level of $L_1$ error, and is more efficient than state-based and edge-based formulations, which validates its effectiveness in practice.

## 5 EXPERIMENTS

We conduct comprehensive experiments to understand the effectiveness of our method and investigate the following key questions: i) How does GAFlowNet compare against previous baselines? ii) What are the effects of state and edge-based flow augmentation, the form of the intrinsic reward mechanism, and critical hyperparameters? iii) Can it scale to larger-scale and more complex tasks?

### 5.1 GRIDWORLD

We first conduct a series of experiments based on GridWorld with sparse rewards (Figure 4). The task is the same as introduced in (Bengio et al., 2021a), except that the reward function is sparse as described in Section 4, which makes it much harder due to the challenge of exploration. With a larger value of the size $H$, it requires the agent to plan in a longer horizon and learn from sparse reward signals. Actions include operations to increase one coordinate as in (Bengio et al., 2021a), and a stop operation indicating termination to guarantee that the underlying MDP is a directed acyclic graph. We compare GAFlowNet against strong baselines including Metropolis-Hastings-MCMC (Dai et al., 2020), PPO (Schulman et al., 2017b), and a GFlowNet (Malkin et al., 2022).

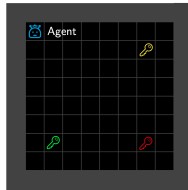

Figure 4: The GridWorld task.

We also include a variant of PPO with intrinsic rewards based on the same intrinsic motivation mechanism using RND (Burda et al., 2018). All baselines are implemented based on the open-source code[1]. Each algorithm is run for five random seeds, and we report their mean and standard deviation. A detailed description of the hyperparameters and setup can be found in Appendix C.1.

#### 5.1.1 PERFORMANCE COMPARISON

We conduct experiments on small, medium, and large GridWorlds with increasing sizes $H$. Full results of other values of $H$ can be found in Appendix C.3. To investigate the effectiveness of GAFlowNet, we first compare it against baselines in terms of the empirical $L_1$ error as computed in Section 4. As shown in the first row in Figure 5, MCMC and PPO fail to converge due to the particularly sparse rewards. Although PPO-RND has a smaller $L_1$ error, it still underperforms GFlowNets by a large margin. GAFlowNet converges fastest and to the smallest level of $L_1$ error, which shows that our method is effective to both explore efficiently and converge to sampling goals with probability proportional to the extrinsic reward function even if the reward signals are sparse.

The number of modes that each method discovers during the course of training is shown in the second row in Figure 5. Although incorporating PPO with intrinsic rewards improves the number of discovered modes compared to that of PPO in larger-scale tasks, it still plateaus quickly. On the other hand, GFlowNets can get trapped in a few modes, while GAFlowNet is able to discover all of the modes efficiently. A detailed comparison in terms of performance can be found in Appendix C.4. Visualization of the learned structures can be found in Appendix C.2. Besides the

---

[1]https://github.com/GFNOrg/gflownet

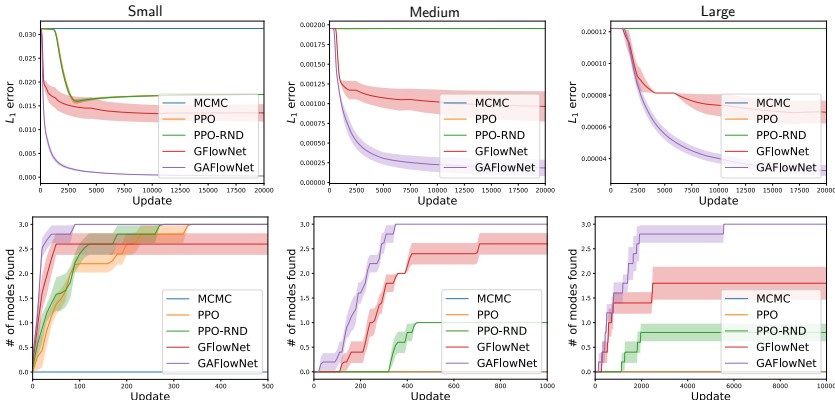

Figure 5: Comparison of GAFlowNets and baselines in GridWorld with increasing sizes corresponding to each column (left: small, middle: medium, right: large). The first and second rows correspond to empirical $L_1$ error and the number of discovered modes, respectively.

particularly challenging sparse reward tasks, we show that GAFlowNets also provide consistent performance improvement in tasks with dense reward functions as in (Bengio et al., 2021a; Malkin et al., 2022). Results and discussions can be found in Appendix C.5 due to space limitations.

### 5.1.2 ABLATION STUDY

We now provide an in-depth ablation study on the important components and hyperparameters of GAFlowNet in the large GridWorld task. We also study the effect of different mechanisms of intrinsic rewards besides RND, where results can be found in Appendix C.6.

**The effect of state-based and edge-based flow augmentation.** In Figure 6(a), we investigate the effect of edge-based, state-based, and joint intrinsic rewards. As discussed in Section 4, incorporating intrinsic rewards for the trajectory in a state-based manner can result in slower convergence, which has a large $L_1$ error. On the other hand, augmenting the TB objective with intrinsic rewards in an edge-based way still fails to motivate the agent to visit states with zero rewards. In contrast, the joint augmentation mechanism is effective in both diversity and performance, achieving the smallest level of $L_1$ error in our experiments. It is also worth noting that only incorporating the intrinsic reward for the terminal state using state-based augmentation is less efficient, which implies the importance of both edge-based and terminal state-based intrinsic rewards.

**The effect of the coefficient of intrinsic rewards.** In practice, we scale intrinsic rewards by a coefficient. Figure 6(b) illustrates the effect of the coefficient of the intrinsic rewards. A too small coefficient does not improve the performance, while a too large coefficient converges slower. There exists an intermediate value that provides the best trade off.

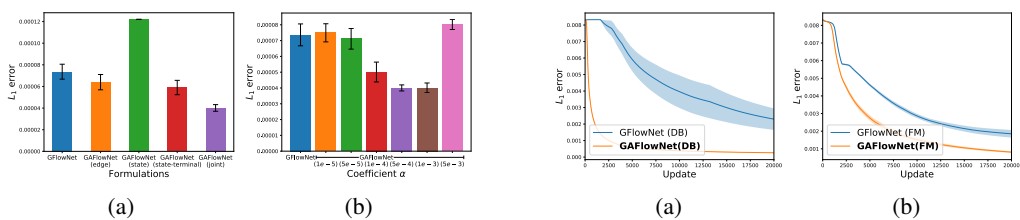

Figure 6: Ablation study. (a) The effect of state- and edge-based intrinsic rewards. (b) The effect of the coefficient of intrinsic rewards.

Figure 7: Empirical $L_1$ error of GFlowNet and GAFlowNet based on (a) DB and (b) FM.

### 5.1.3 VERSATILITY

We now demonstrate that our proposed framework is versatile by building it upon the other two GFlowNet objectives based on the detailed balance (DB) (Bengio et al., 2021b) and flow matching (FM) (Bengio et al., 2021a) criteria. Comparison of empirical $L_1$ error averaged over increasing

sizes $H$ are summarized in Figure 7. As demonstrated, GAFlowNet also significantly improves training convergence of DB and FM, which provides consistent improvement gains.

## 5.2 MOLECULE GENERATION

### 5.2.1 EXPERIMENTAL SETUP

We now investigate the effectiveness of our method in larger-scale tasks, by evaluating it on the more challenging molecule generation task (Bengio et al., 2021a) as depicted in Figure 8(a). A molecule is represented by a graph, which consists of a vocabulary of building blocks. The agent sequentially generates the molecule by choosing where to attach a block and also which block to attach at each step considering chemical validity constraints. There is also an exit action indicating whether the agent decides to stop the generation process. This problem is challenging with large state (about $10^{16}$) and action (around 100 to 2000) spaces. The agent aims to discover diverse molecules with high rewards, *i.e.*, low binding energy to the soluble epoxide hydrolase (sEH) protein. We use a pretrained proxy model to compute this binding energy. We consider a sparse reward function here, where the agent only obtains a non-zero reward if the corresponding molecule succeeds to meet a target score, and the reward is 0 otherwise. A detailed description of the environment is in Appendix C.1.1. We compare our method with previous GFlowNet results (Bengio et al., 2021a), PPO (Schulman et al., 2017b), PPO with intrinsic rewards based on RND, and MARS (Xie et al., 2020). All baselines are run with three random seeds as in (Bengio et al., 2021a). More details for the setup can be found in Appendix C.1.2.

### 5.2.2 PERFORMANCE COMPARISON

We follow the evaluation metric in (Bengio et al., 2021a) and investigate our method in both performance and diversity. Figure 8(b) demonstrates the average reward of the top-10 unique molecules generated by each method. The number of modes discovered by each method with rewards above 7.5 is summarized in Figure 8(c). We compute the average pairwise Tanimoto similarities for the top-10 samples in Figure 8(d). Additional comparison results can be found in Appendix C.7.

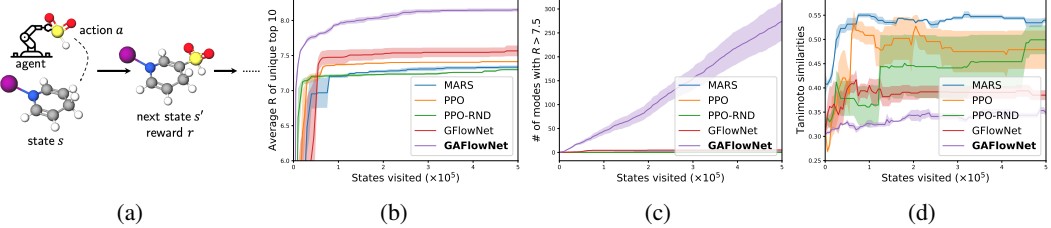

(a)          (b)          (c)          (d)

Figure 8: Molecule generation task. (a) The environment. (b) Average reward of the top-10 molecules. (c) The number of modes with $R > 7.5$. (d) Tanimoto similarity (lower is better).

As shown, MARS fails to perform well given sparse rewards since most of the reward signals are non-informative. On the other hand, PPO and its variant with intrinsic rewards are better at finding higher-quality solutions than MARS, but suffer both from high similarities of the samples. The unaugmented GFlowNet is better at discovering more diverse molecules, but does not perform well in terms of solution quality. GAFlowNet significantly outperforms baseline methods in performance and diversity. We also visualize the top-10 molecules generated by GFlowNet and GAFlowNet in a run in Appendix C.8. As shown, GAFlowNet is able to generate diverse and high-quality molecules efficiently, which demonstrates consistent and significant performance improvement.

## 6 CONCLUSION

In this paper, we propose a new learning framework, GAFlowNet, for GFlowNet to incorporate intermediate rewards. We specify intermediate rewards by intrinsic motivation to tackle the exploration problem of GFlowNets in sparse reward tasks, where it can get trapped in a few modes. We conduct extensive experiments to evaluate the effectiveness of GAFlowNets, which significantly outperforms strong baselines in terms of diversity, convergence, and performance when the rewards are very sparse. GAFlowNet is also scalable to complex tasks like molecular graph generation.

ACKNOWLEDGEMENT

The authors would like to thank Emmanuel Bengio, Qingpeng Cai, and anonymous reviewers for their comments and insightful discussions. Aaron Courville thanks the support of CIFAR. Longbo Huang is supported in part by the Technology and Innovation Major Project of the Ministry of Science and Technology of China under Grant 2020AAA0108400 and 2020AAA0108403, the Tsinghua University Initiative Scientific Research Program, and Tsinghua Precision Medicine Foundation 10001020109. Yoshua Bengio acknowledges the funding from CIFAR, Samsung, IBM and Microsoft.

REPRODUCIBILITY STATEMENT

All details for our experiments are in Appendix C with a detailed description of the task, hyper-parameters, network architectures for baselines, and setup. Our implementation for all baselines and environments is based on open-source repositories. The proof of Theorem 1 can be found in Appendix B. The code is publicly available at `https://github.com/ling-pan/GAFN`.

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

# A DERIVATION OF EDGE-BASED, STATE-BASED, AND JOINT INTERMEDIATE REWARD AUGMENTATION

For the edge-based intermediate reward augmentation, we can obtain the detailed balance objective with the incorporation of intermediate rewards as in Eq. (8) following Section 4.1.

$$F(\mathbf{s}_{t-1})P_F(\mathbf{s}_t|\mathbf{s}_{t-1}) = F(\mathbf{s}_t)P_B(\mathbf{s}_{t-1}|\mathbf{s}_t) + r(\mathbf{s}_{t-1} \to \mathbf{s}_t). \tag{8}$$

Therefore, we have that

$$\begin{cases} t = 1 & F(\mathbf{s}_0)P_F(\mathbf{s}_1|\mathbf{s}_0) = F(\mathbf{s}_1)P_B(\mathbf{s}_0|\mathbf{s}_1) + r(\mathbf{s}_0 \to \mathbf{s}_1) \\ \vdots & \vdots \\ t = n & F(\mathbf{s}_{n-1})P_F(\mathbf{s}_n|\mathbf{s}_{n-1}) = F(\mathbf{s}_n)P_B(\mathbf{s}_n|\mathbf{s}_{n-1}) + r(\mathbf{s}_{n-1} \to \mathbf{s}_n) \end{cases} \tag{9}$$

By accumulative multiplication on both sides, we get that

$$F(\mathbf{s}_0) \cdots F(\mathbf{s}_{n-1}) \prod_{t=0}^{n-1} P_F(\mathbf{s}_{t+1}|\mathbf{s}_t) = F(\mathbf{s}_1) \cdots F(\mathbf{s}_n) \prod_{t=0}^{n-1} \left[ P_B(\mathbf{s}_t|\mathbf{s}_{t+1}) + \frac{r(\mathbf{s}_t \to \mathbf{s}_{t+1})}{F(\mathbf{s}_{t+1})} \right]. \tag{10}$$

Therefore, we obtain the corresponding edge-based reward augmented formulation for trajectory balance as

$$F(\mathbf{s}_0) \prod_{t=0}^{n-1} P_F(\mathbf{s}_{t+1}|\mathbf{s}_t) = F(\mathbf{s}_n) \prod_{t=0}^{n-1} \left[ P_B(\mathbf{s}_t|\mathbf{s}_{t+1}) + \frac{r(\mathbf{s}_t \to \mathbf{s}_{t+1})}{F(\mathbf{s}_{t+1})} \right], \tag{11}$$

where $F(\mathbf{s}_0) = Z = \sum_{\mathbf{x}} R(\mathbf{x}) + \sum_{\mathbf{s}_{t-1} \to \mathbf{s}_t} r(\mathbf{s}_{t-1} \to \mathbf{s}_t)$, and $F(\mathbf{s}_n) = R(\mathbf{x})$.

The state-based reward augmented formulation can be obtained similarly by following Appendix D in (Bengio et al., 2021b). The joint reward augmented formulation is obtained by combining edge-based and state-based reward augmentation.

# B PROOF OF THEOREM 1

**Theorem 1.** *Suppose that $\forall \tau, \mathcal{L}_{\text{GAFlowNet}}(\tau) = 0$, and $\forall \mathbf{x}, R(\mathbf{x}) + r(\mathbf{x}) > 0$. When edge-based intrinsic rewards converge to 0, we have that (1) $P(\mathbf{x}) = \frac{R(\mathbf{x})+r(\mathbf{x})}{\sum_{\mathbf{x}}[R(\mathbf{x})+r(\mathbf{x})]}$; (2) If state-based intrinsic rewards converge to 0, then $P(\mathbf{x})$ is an unbiased sample distribution.*

*Proof.* By definition, we have that

$$F_\theta(\tau) = Z \prod_{t=0}^{n-1} P_F(\mathbf{s}_{t+1}|\mathbf{s}_t). \tag{12}$$

Since $\forall \tau, \mathcal{L}_{\text{GAFlowNet}}(\tau) = 0$, we have that

$$Z \prod_{t=0}^{n-1} P_F(\mathbf{s}_{t+1}|\mathbf{s}_t) = (R(\mathbf{x}) + r(\mathbf{x})) \prod_{t=0}^{n-1} \left[ P_B(\mathbf{s}_t|\mathbf{s}_{t+1}) + \frac{r(\mathbf{s}_t \to \mathbf{s}_{t+1})}{F(\mathbf{s}_{t+1})} \right] \tag{13}$$

Therefore, we obtain that

$$P_\theta(\tau) = \frac{F_\theta(\tau)}{Z} = \frac{R(\mathbf{x}) + r(\mathbf{x})}{Z} \prod_{t=0}^{n-1} \left[ P_B(\mathbf{s}_t|\mathbf{s}_{t+1}) + \frac{r(\mathbf{s}_t \to \mathbf{s}_{t+1})}{F(\mathbf{s}_{t+1})} \right]. \tag{14}$$

When edge-based intrinsic rewards converge to 0 and $F$ does not vanish, we have that

$$P_\theta(\mathbf{x}) = \sum_{\tau=(\mathbf{s}_0 \to \cdots \to \mathbf{s}_n = \mathbf{x})} P_\theta(\tau) = \frac{R(\mathbf{x}) + r(\mathbf{x})}{Z} \sum_{\tau=(\mathbf{s}_0 \to \cdots \to \mathbf{s}_n = \mathbf{x})} \prod_{t=0}^{n-1} P_B(\mathbf{s}_t|\mathbf{s}_{t+1}). \tag{15}$$

Due to the law of total probability, we have that

$$\sum_{\tau=(\mathbf{s}_0\rightarrow\cdots\rightarrow\mathbf{s}_n=\mathbf{x})}\prod_{t=0}^{n-1}P_B(\mathbf{s}_t|\mathbf{s}_{t+1}) = 1. \tag{16}$$

Therefore, $P_\theta(\mathbf{x}) = \frac{R(\mathbf{x})+r(\mathbf{x})}{Z}$. As $\sum_x P_\theta(\mathbf{x}) = 1$, we also get that $Z = \sum_{\mathbf{x}}(R(\mathbf{x})+r(\mathbf{x}))$.

Therefore, we have Part (1) that

$$P_\theta(\mathbf{x}) = \frac{R(\mathbf{x})+r(\mathbf{x})}{\sum_{\mathbf{x}}[R(\mathbf{x})+r(\mathbf{x})]} \tag{17}$$

Based on the above analysis, $P(\mathbf{x})$ is an unbiased estimation when state-based intrinsic rewards converge to 0, and we have Part (2).

$\square$

## C  EXPERIMENTAL DETAILS

### C.1  EXPERIMENTAL SETUP

#### C.1.1  TASK

**The molecule generation task**   We adopt a pretrained proxy model for the reward, which is trained on a dataset of $300,000$ molecules that are randomly generated as provided in (Bengio et al., 2021a). For the original dense reward function, the agent receives a reward based on the normalized score. Here, we use a sparse reward function, where the agent only obtains the original non-zero reward if the normalized score succeeds to meet a target score $(7.0)$, and the reward is 0 otherwise. As described in Section 5.2.1, the agent can choose one of the blocks to attach from the basic building blocks vocabulary (with a size of 105).

#### C.1.2  BASELINE

All baseline methods are implemented based on the open-source implementation as described in the main text, where we follow the default hyperparameters and setup as in (Bengio et al., 2021a). The code will be released upon publication of the paper.

**GridWorld**   Specifically, in GridWorld, the GFlowNet model is a feedforward network consisting of two hidden layers with 256 hidden units per layer using LeakyReLU activation. We train all models based on samples from a parallel of 16 rollouts in the environment. We leverage random network distillation (RND) (Burda et al., 2018) as the intrinsic reward mechanism, where the random target network and the predictor network are both feedforward networks consisting of two hidden layers with 256 hidden units per layer using LeakyReLU activation. We train the GFlowNet model and RND jointly based on the Adam (Kingma & Ba, 2014) optimizer with a learning rate of 0.001 for the policy models ($P_F$ and $P_B$) and 0.1 for $Z$.

**Molecule generation**   For the molecule generation task, we use a reward proxy provided in (Bengio et al., 2021a). As the molecule is represented as an atom graph, we use Message Passing Neural Networks (MPNN) (Gilmer et al., 2017) as the network architecture for all models. Note that we build our method upon GFlowNet based on the flow matching criterion in the molecule generation task, since it is the most competitive version in this task in terms of finding high-quality and diverse candidates.

For GAFlowNet, the only hyperparameter that requires tuning is the coefficient $\alpha$ of intrinsic rewards, where we use a same value for state-based and edge-based augmentation. We tune $\alpha$ in $\{0.001, 0.005, 0.01, 0.05, 0.1, 0.5\}$ with grid search. Specifically, $\alpha = 0.001$ for GridWorld with all values of horizon except for $H \in \{16, 64\}$, where we set $\alpha$ to be 0.005. For the molecule generation task, $\alpha$ is set to be 0.1.

## C.2   VISUALIZATION OF THE LEARNED STRUCTURES

In this section, we visualize the learned forward policies of GFlowNets and GAFlowNets. The results are demonstrated in Figure 9 in the small GridWorld, and the borders (the last column and the last row) of the grid are omitted in the figure for better readability. The length of the arrows is proportional to the likelihood of the corresponding actions under the forward policy $P_F$, while the size of the gray circle is proportional to the probability of the termination action (stop). As shown, GFlowNet can get trapped in a few modes, while GAFlowNets is able to discover all the modes.

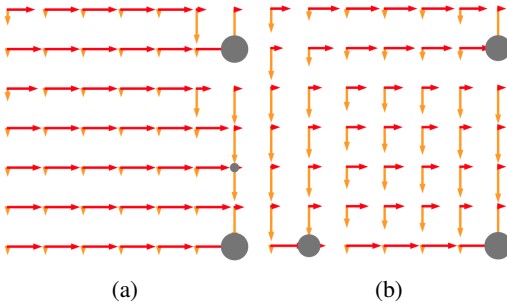

(a)            (b)

Figure 9: Visualization of the learned forward policies of (a) GFlowNets and (b) GAFlowNets in GridWorld (horizon $H = 8$).

The corresponding distribution of the samples collected by GFlowNets and GAFlowNets during the course of training in the small GridWorld task is shown in Figure 10, where GAFlowNets leads to more diverse solutions. Visualization of the top molecules discovered by GFlowNets and GAFlowNets can be found in Appendix C.8.

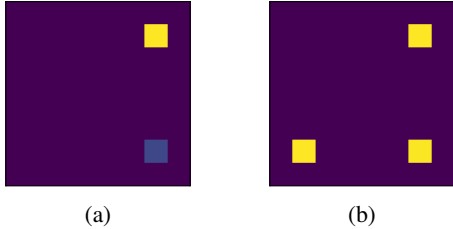

(a)            (b)

Figure 10: Distribution of the samples during the course of training from (a) GFlowNets and (b) GAFlowNets in GridWorld (horizon $H = 8$).

## C.3   FULL RESULTS IN GRIDWORLD

We show in Figure 11 the full comparison results in GridWorld with increasing sizes $H\{8, 16, 32, 64, 128\}$. As shown, GAFlowNet significantly outperforms baselines in empirical $L_1$ error and the number of modes found.

## C.4   PERFORMANCE COMPARISON

Apart from evaluating our method based on the metrics (the number of modes discovered by each method and empirical $L_1$ error) as in (Bengio et al., 2021a), we are also interested in its performance after each update. Here, we evaluate the performance of baselines after each update (instead of throughout the training process) as in the evaluation scheme of RL algorithms. Figure 12 demonstrates the performance for the top-5 solutions among a batch of 16 parallel rollouts of each method after each update for GridWorld with sizes $H \in \{8, 16, 32, 64, 128\}$. As shown, although PPO is more efficient than PPO, both of them underperform GFlowNet by a large margin (especially with

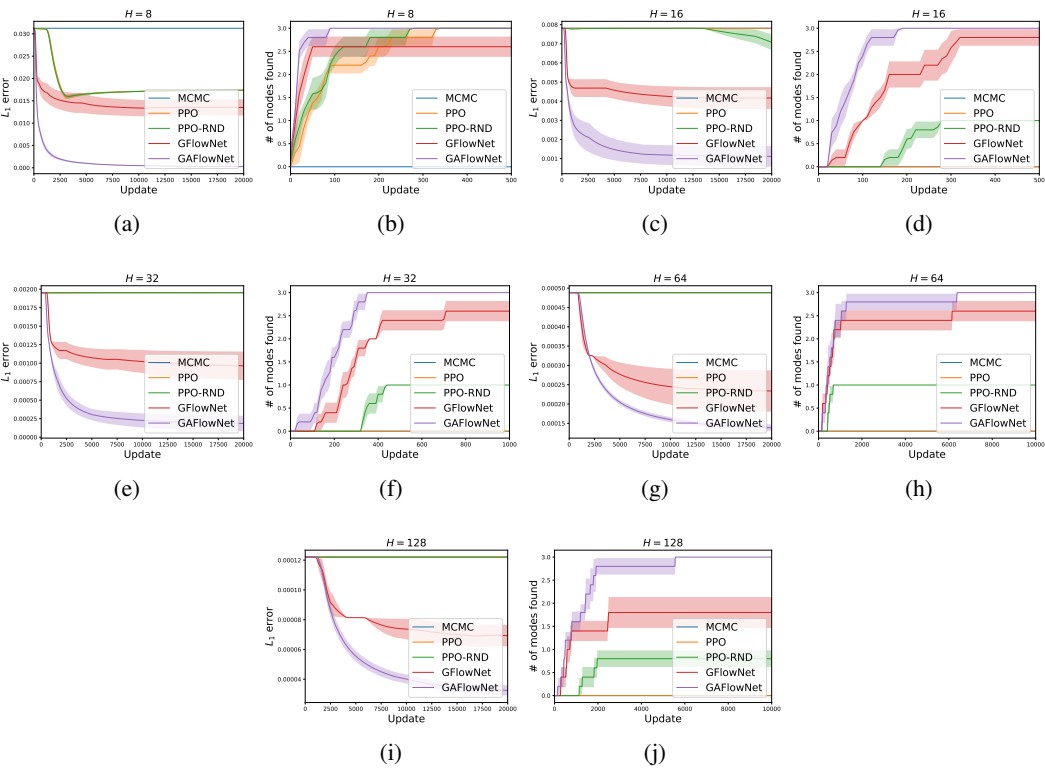

Figure 11: Comparison of GAFlowNets and baselines in GridWorld with increasing sizes $H \in \{8, 16, 32, 64, 128\}$. (a), (c), (e), (g), (i) correspond to the empirical $L_1$ error. (b), (d), (f), (h), (j) correspond to the number of modes discovered by each method.

a larger value of $H$). We find that GAFlowNet significantly outperforms baseline methods, and also performs more efficiently than GFlowNet.

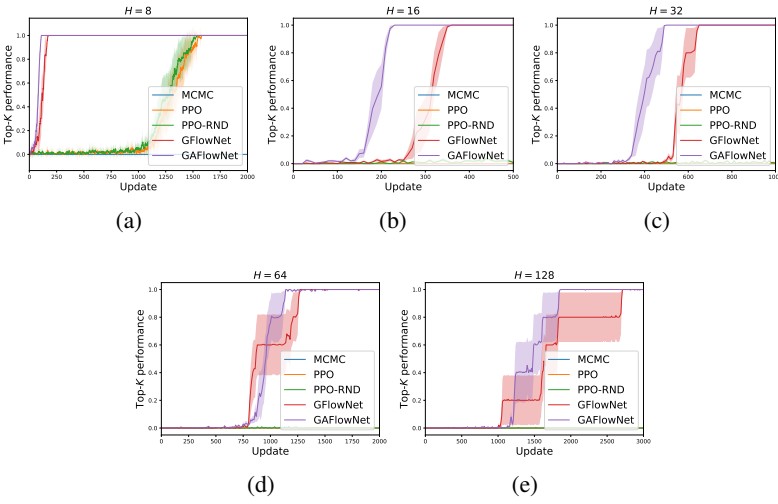

Figure 12: Top-$K$ performance of baselines after each update for horizon $H \in \{8, 16, 32, 64, 128\}$.

## C.5 ADDITIONAL RESULTS ON GRIDWORLDS WITH DENSE REWARDS

In this section, we evaluate the performance of GAFlowNets in a variant of GridWorld with dense reward functions. We employ the same reward function (Eq. (18)) as in (Bengio et al., 2021a; Malkin et al., 2022) with $R_0 = 1e - 3$ (the hardest variant of the grid considered in (Bengio et al., 2021a; Malkin et al., 2022)), which is shown in Figure 13(a). Note that when $R_0$ is closer to 0, the problem becomes more difficult as it results in a region of the state space that is undesirable to explore (Bengio et al., 2021a).

$$R(\mathbf{x}) = R_0 + \frac{1}{2} \prod_i \mathbb{I}\left(0.25 < |x_i/H - 0.5|\right) + 2 \prod_i \mathbb{I}\left(0.3 < |x_i/H - 0.5| < 0.4\right) \qquad (18)$$

Comparison results of GAFlowNets and GFlowNets in empirical $L_1$ error in GridWorlds with different sizes $H \in \{8, 16, 32, 64, 128\}$ are shown in Figures 13(b)-(f). As demonstrated, GAFlowNets also greatly improve the learning efficiency of GFlowNets. This is because GAFlowNets exhibit greater exploration ability, and are more efficient in choosing training trajectories that the agent is not very familiar with yet and may have a high reward (Bengio et al., 2021b). Therefore, it achieves consistent performance improvement.

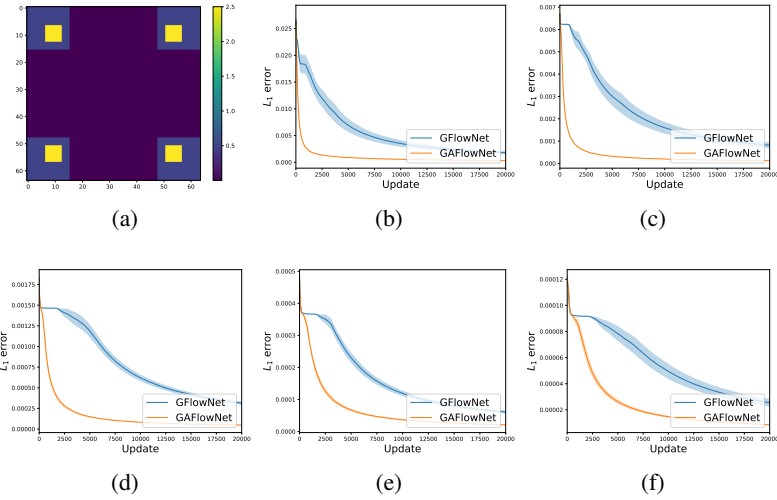

Figure 13: (a) The dense reward function. (b)-(f) Empirical $L_1$ error of GAFlowNets and GFlowNets in GridWorld with dense reward functions for horizon $H \in \{8, 16, 32, 64, 128\}$.

## C.6 ADDITIONAL ABLATION STUDY OF GAFLOWNET

We investigate the effect of different types of intrinsic rewards including Intrinsic Curiosity Module (ICM) (Pathak et al., 2017), Novelty Difference (NovelD) (Zhang et al., 2021), and Random Network Distillation (RND) (Burda et al., 2018) in Figure 14, with fine-tuned coefficients for intrinsic rewards. We also include a baseline with constant intrinsic rewards in GAFlowNet, which mimics the behavior of $\epsilon$-greedy exploration typically used in reinforcement learning algorithms. As demonstrated, GAFlowNet is not sensitive to the forms of intrinsic rewards, but RND enables the fastest convergence in our simulations. It also validates the effectiveness of the novelty-based methods from the comparison of GAFlowNet and GAFlowNet with constant intrinsic rewards. This is because novelty-based methods are more efficient than blindly wandering in the maze.

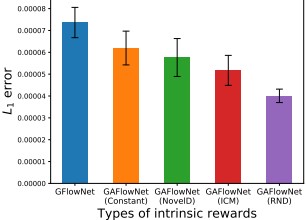

Figure 14: Ablation study on different types of intrinsic rewards.

## C.7 Additional Performance Comparison on the Molecule Generation Task

Following the evaluation metrics in (Bengio et al., 2021a), besides the results in Figure 8 in the main text, we also evaluate the average reward of the top-100 molecules and the number of modes with $R > 8.0$ discovered by each method. As demonstrated in Figure 15, GAFlowNet achieves consistent and significant performance improvement over previous baselines.

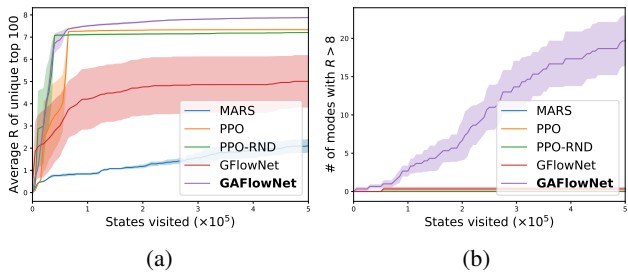

(a)  (b)

Figure 15: Molecule generation task. (a) The average reward of the top-100 molecules. (b) The number of modes with $R > 8.0$.

## C.8 Full Visualization of top-10 molecules

Figure 16 demonstrates the top-10 molecules generated by GFlowNet and GAFlowNet, where GAFlowNet discovers more diverse and higher-quality solutions.

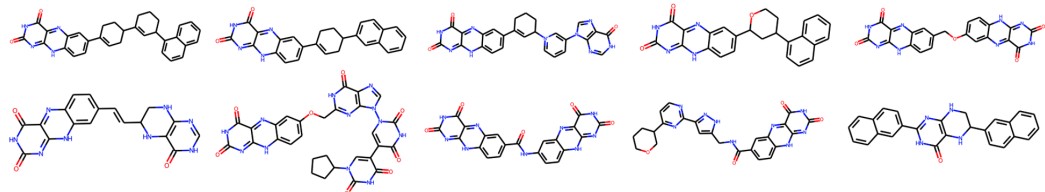

(a) GFlowNet. Tanimoto similarity is $0.410$ and the average reward is $7.747$.

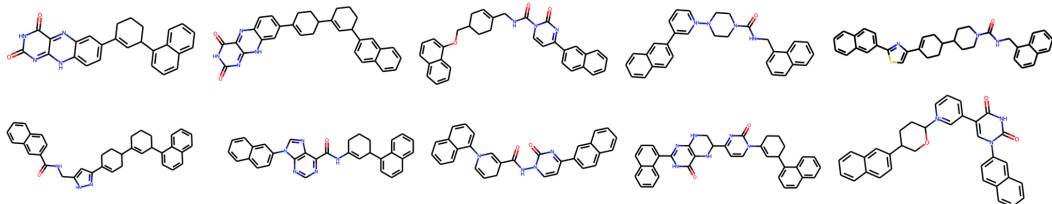

(b) GAFlowNet. Tanimoto similarity is $0.356$ and the average reward is $8.120$.

Figure 16: Full visualization of top-10 molecules generated by GFlowNet and GAFlowNet.

