# OpenReview forum: "Generative Augmented Flow Networks"
_ICLR.cc/2023/Conference — ICLR 2023 notable top 25%_

### Official Review · Reviewer_ka6N · 2022-10-22

**Confidence:** 3
**Correctness:** 3
**Technical Novelty And Significance:** 3
**Empirical Novelty And Significance:** 3
**Recommendation:** 6

**Clarity, Quality, Novelty And Reproducibility:**


* Clarity: In general, the paper is well-written except a few points I raised above (see weaknesses).

* Quality: The paper provides compelling experiments against the related prior approach (GEFlowNets). The comparison to standard RL algorithms (e.g., PPO) could be a bit misleading without presenting results on standard RL benchmarks.

* Novelty: Although the overall method builds upon GEFlowNets, the incorporation of intermediate rewards is novel.

* Reproducibility: The paper provided some details in the appendix and proposed to release the code upon publication.

**Strength And Weaknesses:**

[Strength]

* The proposed idea of incorporating intermediate rewards to GEFlowNets is novel.
* The result convincingly demonstrates that the proposed method significantly outperforms the previous approach (GEFlowNets) both in terms of performance and the diversity of solutions.
* The paper is well-written.

[Weaknesses]

* **Lack of result on standard RL benchmark:** To my understanding, the proposed method is general enough to be applied to any RL problems as it can incorporate intermediate rewards, and the introduction was also written in such a way. However, all of the experiments were conducted on the "reward-at-termination" types of environments, while intermediate rewards were given by intrinsic rewards (not from the environment). It would be much more comprehensive to evaluate the proposed method on a standard RL benchmark (e.g., Atari) that contain "intermediate extrinsic" rewards.

* **Lack of clarity about intermediate rewards / intrinsic rewards:** Related to the above point, I felt that the paper often assumes that intermediate rewards = intrinsic rewards without a clear distinction, which I found confusing. In page 5, for example, it says "our edge-based reward augmentation could motivate the agent to discover other paths ...". Does the diversity of the solution come from the incorporation of intermediate rewards (i.e., the edge-based augmentation) or the combination of the edge-based augmentation and a particular intrinsic rewards designed for exploration? I believe that the paper meant the latter. I would be good to make this distinction clear throughout the paper.

* **Lack of clarity about edge-based augmentation vs state-based augmentation:** I found it difficult to understand the different behaviors between the edge-based augmentation and the state-based augmentation. Why does the edge-based augmentation tend to explore "locally" while the state-based augmentation explores more "globally"? I could not figure it out from the equations immediately. Further clarifying this point would be helpful for justifying the joint approach.

**Summary Of The Paper:**

This paper proposes an extension of Generative Flow Networks (GEFlowNets). The main idea is to incorporate intermediate rewards as well as terminal rewards into the objective, which was missing in GEFlowNets. The paper proposes to combine two variants: edge-based augmentation and state-based augmentation and shows that adding intrinsic rewards leads to the same solution when intrinsic rewards converge to 0 over time. The empirical results on a grid world and a molecule generation task show that the proposed method significantly outperforms GEFlowNets and other RL baselines including PPO when combined with RND-like intrinsic rewards. Besides, it discovers more diverse set of solutions compared to the previous approaches thanks to the additional intrinsic rewards.

**Summary Of The Review:**

GEFlowNets are an interesting alternative to existing RL algorithms, which has been receiving attention. This paper extends GEFlowNets nicely to support intermediate rewards, which sounds like an exciting direction towards solving RL problems with GEFlowNets. Although the results are limited to the intrinsic rewards on specific types of problems, the comparison to the prior work GEFlowNets looks quite promising and convincing. I am willing to increase the score if the authors reflect (or provide a reasonable response to) some of my comments above.

---

> ### Author Response · Authors · 2022-11-11
> **Response to Reviewer ka6N (Part 1 of 2)**
>
> We thank the reviewer for the helpful feedback and detailed evaluation of our work, and we greatly appreciate the comments! Below, we seek to address each of your concerns. We have updated the paper to address the reviewer's comments, where the revisions are marked in blue.
>
> *Q1: It would be much more comprehensive to evaluate the proposed method on a standard RL benchmark (e.g., Atari) that contain "intermediate extrinsic" rewards.*
>
> We thank the reviewer for the insightful comment. Extending our methods to Atari and evaluating it is a very interesting future direction. We believe that our work may open the door to the general RL setting (by considering intermediate rewards) and we are actively working towards this direction.
>
> Regarding your concern, we include an additional experiment as an illustrating example comparing our method against PPO in a GridWorld with intermediate extrinsic rewards (a variant of the GridWorld in [1]) in Appendix C.7. In such a task with dense reward function, we use the extrinsic intermediate reward given from the environment for $r(s_t \to s_{t+1})$ and do not use intrinsic rewards (since it is not a hard exploration game). Results show that GAFlowNets outperforms baselines including PPO, demonstrating the potential of GAFlowNets.
>
> [1] Hong, Z. W., Shann, T. Y., Su, S. Y., Chang, Y. H., Fu, T. J., & Lee, C. Y. (2018). Diversity-driven exploration strategy for deep reinforcement learning. Advances in neural information processing systems, 31.
>
> *Q2: I felt that the paper often assumes that intermediate rewards = intrinsic rewards without a clear distinction, which I found confusing. In page 5, for example, it says "our edge-based reward augmentation could motivate the agent to discover other paths ...". Does the diversity of the solution come from the incorporation of intermediate rewards (i.e., the edge-based augmentation) or the combination of the edge-based augmentation and a particular intrinsic reward designed for exploration? I believe that the paper meant the latter.*
>
> Thank you for the point. We would like to clarify that we specify intrinsic motivation as intermediate rewards to tackle the exploration problem in sparse reward environments (where most of the extrinsic intermediate rewards are zero and the agent only obtains a return at the terminal state). For the second question, yes, the reviewer is correct--the major performance improvement comes from the combination of our augmentation form of intermediate rewards and the intrinsic motivation mechanism for driving the agent to explore novel regions. We clarified this point in the paper and will make sure to update the paper to make it more clear and avoid confusion.

---

> > ### Author Response · Authors · 2022-11-11
> > **Response to Reviewer ka6N (Part 2)**
> >
> > *Q3: Why does the edge-based augmentation tend to explore "locally" while the state-based augmentation explores more "globally"?*
> >
> > By "global" exploration, we mean that it is able to motivate the agent to explore novel terminal states (explore the solution space globally). From Eq. (5), we see that for the state-based approach, the augmented return for $\textbf{x}$ turns to $R(\textbf{x}) + \sum_{t=0}^{n-1} r(s_t \to s_{t+1})$, which is explicitly changed. According to the learning mechanism of GFlowNets, the probability mass put on $\textbf{x}$ will be proportional to this return. As a result, the agent will be directly motivated to explore novel terminal states even with zero rewards, which can be beneficial for discovering other interesting novel solutions and leads to more global exploration. On the other hand, for the edge-based approach in Eq. (4), it motivates the agent to explore novel transitions. However, it does not guarantee to visit novel terminal states with zero rewards, which results in exploration to a more local degree. We will make sure to clarify this point in the paper.
> >
> > *Q4: The comparison to standard RL algorithms (e.g., PPO) could be a bit misleading without presenting results on standard RL benchmarks.*
> >
> > Our experimental design (benchmark and baselines) follows previous works of GFlowNets [2, 3, 4] and evaluates our method and baselines (e.g., PPO) on tasks that are already used to benchmark GFlowNets. PPO has also been used for generating molecular graphs [5], and is included for comparison in molecule generation and gridworlds here for completeness (to verify that GFlowNets-based methods can discover more modes than PPO that focuses on a single mode of the reward function). In addition, we also include an additional comparison to PPO in tasks with extrinsic intermediate rewards in Appendix C.7, where GAFlowNet also outperforms PPO. We hope that the explanation could address your concern regarding the comparison of PPO.
> >
> > [2] Bengio, E., Jain, M., Korablyov, M., Precup, D., & Bengio, Y. (2021). Flow network based generative models for non-iterative diverse candidate generation. Advances in Neural Information Processing Systems, 34, 27381-27394.
> >
> > [3] ​​Malkin, N., Jain, M., Bengio, E., Sun, C., & Bengio, Y. (2022). Trajectory Balance: Improved Credit Assignment in GFlowNets. Advances in Neural Information Processing Systems, 35.
> >
> > [4] Bengio, Y., Deleu, T., Hu, E. J., Lahlou, S., Tiwari, M., & Bengio, E. (2021). Gflownet foundations. arXiv preprint arXiv:2111.09266.
> >
> > [5] You, J., Liu, B., Ying, Z., Pande, V., & Leskovec, J. (2018). Graph convolutional policy network for goal-directed molecular graph generation. Advances in neural information processing systems, 31.
> >
> > We thank the reviewer for the time and effort in reviewing our work! We would very much appreciate it if the reviewer can check our responses and the updates in the paper and let us know whether they address your concerns. We are happy to provide further clarification if you have any additional concerns.

---

### Official Review · Reviewer_kXmH · 2022-10-23

**Confidence:** 4
**Clarity, Quality, Novelty And Reproducibility:** The novelty is limited.
**Correctness:** 2
**Technical Novelty And Significance:** 2
**Empirical Novelty And Significance:** 2
**Recommendation:** 6

**Strength And Weaknesses:**

Strength:

(1)The authors focus on the intermediate rewards in the learning of Generative Flow Network, and propose a novel GFlowNet learning framework, dubbed GAFlowNet, to incorporate intermediate rewards.

 (2) The paper proposes three different kinds of augmentation methods, namely, edge-based intermediate reward augmentation, state-based intermediate reward augmentation, and joint intermediate reward augmentation. And, the advantages and disadvantages of different augmentation methods are analyzed intuitively and clearly.

 (3) In general, the experiments are comprehensive and reasonable.

 (4) In general, the paper is well-organized and easy to follow.

Weakness:

(1)  The author just considers combining intermediate rewards to further improve the performance of GFlowNet in particularly challenging sparse reward tasks, which is not novel enough.

(2) More novelty degree measure methods can be considered to further refine the experiments.

Minor errors:
(1) In Figure.5, the first and second rows correspond to the number of discovered modes and empirical L1 error respectively, which doesn’t correspond to the descriptions in the caption of Figure.5 and the analysis of the experiment in 5.1.1.

(2) The position of the figures can be adjusted to make the paper read more fluently, for example, Figure.4 has been mentioned earlier in the paper.


**Summary Of The Paper:**

The paper argues that intermediate rewards are also crucial in the learning of Generative Flow Network, and proposes a novel GFlowNet learning framework, dubbed GAFlowNet, to incorporate intermediate rewards. The paper specifies intermediate rewards by intrinsic motivation to deal with the exploration of state space for GFlowNets in sparse reward tasks. The experiment results on the GridWorld domain and the larger-scale and more challenging molecule generation task show the effectiveness of the method in terms of convergence, diversity, and performance.

**Summary Of The Review:**

Overall, this paper proposes an interesting method. But, why intermediate rewards play a critical role in RL  is not clear.  It is unclear that why the improved Generative Augmented Flow Networks (GAFlowNets) can work better than the previous Generative Flow Network.


The rebuttal is good, and I thus increase the score.

---

> ### Author Response · Authors · 2022-11-11
> **Response to Reviewer kXmH (Part 1 of 2)**
>
> We thank the reviewer for the reading and the detailed feedback of our work! Below, we seek to address each of your concerns. We have updated the paper to address the reviewer's comments, where the revisions are marked in blue.
>
> *Q1: The author just considers combining intermediate rewards to further improve the performance of GFlowNet in particularly challenging sparse reward tasks, which is not novel enough.*
>
> Regarding the novelty concern, we would like to emphasize that we propose a novel learning framework to incorporate intermediate rewards to GFlowNets besides the significant performance improvement. Note that the current mathematical framework of GFlowNets only allows for terminal rewards (as events that happen only once per trajectory) unlike the standard reinforcement learning frameworks, and aims to solve the sampling problem. Considering intermediate rewards into the learning of GFlowNets sheds light on combining general reinforcement learning and classical inference approaches, which is a non-trivial contribution and has the potential to motivate fruitful future directions between the two fields.
>
> In addition, sparse reward tasks are common in real-world applications, and how to tackle the sparsity of rewards has been known as a very challenging problem, since the agent can fail to explore efficiently. Regarding your concern, we also include additional experiments of GAFlowNets in GridWorlds with dense reward functions (the same one as in [1, 2]) in Appendix C.5. Specifically, the reward function is $R(\textbf{x}) = R_0 + \frac{1}{2} \prod_i \mathbb{I}\left(0.25<\left|x_i / H-0.5\right|\right) + 2 \prod_i \mathbb{I}\left(0.3<\left|x_i / H-0.5\right|<0.4\right)$, where $R_0=1e-3$ (the hardest variant of reward functions used in [1, 2]). As shown in Figure 14 in Appendix C.5, GAFlowNets also learns much more efficiently than GFlowNets. This is because it exhibits great exploration ability, and can drive the agent to visit regions that it is not familiar with and may have a high reward.
>
> [1] Bengio, E., Jain, M., Korablyov, M., Precup, D., & Bengio, Y. (2021). Flow network based generative models for non-iterative diverse candidate generation. Advances in Neural Information Processing Systems, 34, 27381-27394.
>
> [2] ​​Malkin, N., Jain, M., Bengio, E., Sun, C., & Bengio, Y. (2022). Trajectory Balance: Improved Credit Assignment in GFlowNets. Advances in Neural Information Processing Systems, 35.
>
> *Q2: More novelty degree measure methods can be considered to further refine the experiments.*
>
> We have investigated the effect of recent novelty degree measure methods including RND [3], ICM [4], and NovelD [5], in Appendix C.6. Results show that they are all effective in improving the learning of GFlowNets due to their improved exploration ability, where RND leads to a smallest empirical $L_1$ error. Results verify that GAFlowNet is a general formulation and can consider different types of intrinsic rewards.
>
> [3] Burda, Y., Edwards, H., Storkey, A., & Klimov, O. (2018). Exploration by random network distillation. In International Conference on Learning Representations.
>
> [4] Pathak, D., Agrawal, P., Efros, A. A., & Darrell, T. (2017). Curiosity-driven exploration by self-supervised prediction. In International conference on machine learning (pp. 2778-2787). PMLR.
>
> [5] Zhang, T., Xu, H., Wang, X., Wu, Y., Keutzer, K., Gonzalez, J. E., & Tian, Y. (2021). Noveld: A simple yet effective exploration criterion. Advances in Neural Information Processing Systems, 34, 25217-25230.

---

> > ### Author Response · Authors · 2022-11-11
> > **Response to Reviewer kXmH (Part 2)**
> >
> > *Q3: Why intermediate rewards play a critical role in RL is not clear.*
> >
> > Intermediate rewards provide intermediate feedback and learning signals to the agent at each step to help it accomplish its task. For example, as discussed in [6], learning can be difficult without intermediate feedback for RL agents.
> >
> > [6] Silver, D., Singh, S., Precup, D., & Sutton, R. S. Reward Is Enough. Artificial Intelligence.
> >
> > *Q4: It is unclear that why the improved Generative Augmented Flow Networks (GAFlowNets) can work better than the previous Generative Flow Network.*
> >
> > It has been shown that GFlowNets can get trapped in a few modes [7], which is a very serious but common issue for probabilistic inference problems [8]. The reason why GAFlowNets outperform GFlowNets is that it motivates the agent to explore novel states that have not been visited yet and may have a high reward. This is achieved by providing intermediate feedback considering intrinsic motivation to the agent at each step, which explicitly drives the agent to escape from local modes.
> >
> > [7] Jain, M., Bengio, E., Hernandez-Garcia, A., Rector-Brooks, J., Dossou, B. F., Ekbote, C. A., ... & Bengio, Y. (2022, June). Biological sequence design with gflownets. In International Conference on Machine Learning (pp. 9786-9801). PMLR.
> >
> > [8] Minka, T. (2005). Divergence measures and message passing. Technical report, Microsoft Research.
> >
> > *Minor points: Captions in the figure and positions of the figures.*
> >
> > Thank you for pointing these out. We have exchanged the first and second rows of Figure 5 and will update the figure in the paper.
> >
> > We thank the reviewer for the time and effort in reviewing our work! We would very much appreciate it if the reviewer can check our responses and the updates in the paper and let us know whether they address your concerns. We are happy to provide further clarification if you have any additional concerns.

---

> > ### Comment · Reviewer_kXmH · 2022-11-16
> > **Why incorporating intermediate rewards to GFlowNets can bring  significant performance improvement?**
> >
> > The current mathematical framework of GFlowNets only allows for terminal rewards, so you consider the intermediate rewards. This cannot be a reason. It needs more evidence to state/prove the improvement.
> >
> >
> > "Considering intermediate rewards into the learning of GFlowNets sheds light on combining general reinforcement learning and classical inference approaches, which is a non-trivial contribution and has the potential to motivate fruitful future directions between the two fields."
> >
> > The previous GFlowNets do not consider the intermediate rewards, but you consider them. It cannot be a non-trivial contribution unless you can prove or present some strong reasons.
> >
> > From optimization, it likes you introduce more signals to push a fast convergence. You need some bounds or analysis to show that this new convergence manner can work better than the previous one signal variance.  Just like the typical muti-variable optimization or optimization decomposition.
> >
> > If we follow your idea, there are more questions: 1) when the intermediate reward's quality is low/poor, what will happen? 2) how many intermediate rewards are reasonable? 3) will n intermediate rewards be better than m if n>m?....
> >
> > Learning with more supervision may not be so positive or beneficial.
> >
> > I will keep my score until the authors can relieve or solve my puzzle.
> >
> > Considering that this work presents a new learning manner/style, I also will consider increasing the score even if it lacks some strong logic. It depends on the subsequent response. Thanks~

---

> > > ### Author Response · Authors · 2022-11-17
> > > **Response to Reviewer kXmH (Part 1 of 2)**
> > >
> > > We thank the reviewer for the reply, and for the time and effort in reviewing our paper! We seek to address each of your concerns below.
> > >
> > > *Q1: The previous GFlowNets do not consider the intermediate rewards, but you consider them. It needs more evidence to state/prove the improvement, and cannot be a non-trivial contribution unless you can prove or present some strong reasons.*
> > >
> > > A major advantage of introducing intermediate rewards to the learning framework of GFlowNets is that it can help the agent to efficiently explore even in the absence of informative extrinsic rewards through intrinsic motivation (formulated as intrinsic intermediate rewards) at each step. This provides a dense guidance reward to the agent for discovering novel states. In addition, it can help the agent to prevent getting trapped in a few modes in hard exploration tasks, which is a critical challenge for GFlowNets [1, 2, 3].
> > >
> > > Consider the GridWorld task, where the agent only receives a reward of $+1$ when it reaches one of the three goal positions, and the reward is $0$ otherwise. The sparsity of informative reward feedback poses a critical challenge to the agent for efficient exploration -- the agent gets a reward of $0$ when it wanders around, hits the wall or reaches non-goal positions. Therefore, it can be difficult for the agent to sequentially collect a successful trajectory for meaningful updates of the model. In addition, it can get trapped in a few modes in hard exploration tasks as shown in the paper (Figures 1 and 5).
> > >
> > > In psychology, intrinsic motivation has been shown to be powerful for agents to explore efficiently driven by curiosity [4, 5]. For example, RND [7] is a popular intrinsic motivation-based method, which outputs a large exploration bonus for novel/unfamiliar states and a small incentive for known/familiar states at each step, which are adaptively adjusted during learning. Therefore, based on this kind of dense intermediate reward signal according to the novelty of states at each step, it helps the agent to discover new knowledge about the world more efficiently.
> > >
> > > However, such ideas have not been explored with GFlowNets because its current mathematical framework only allows for terminal rewards [10], and does not take (intrinsic) intermediate rewards into account. This deficiency as well as the potential of introducing intrinsic intermediate rewards based on curiosity to tackle hard exploration tasks is one of the motivations of this paper.
> > >
> > > *Q2: From optimization, it looks like you introduce more signals to push a fast convergence. You need some bounds or analysis to show that this new convergence manner can work better than the previous one signal variance. Just like the typical muti-variable optimization or optimization decomposition.*
> > >
> > > Thanks for the suggestion. To the best of our knowledge, the analysis of the convergence rate with non-linear function approximators as used in our work is challenging and remains an open problem for GFlowNets. It is an interesting direction for analyzing the convergence rate of GAFlowNets. This paper focuses on algorithmic development and we demonstrate the effectiveness of our algorithm via extensive experiments.

---

> > > > ### Author Response · Authors · 2022-11-17
> > > > **Response to Reviewer kXmH (Part 2)**
> > > >
> > > > *Q3: If we follow your idea, there are more questions: 1) when the intermediate reward's quality is low/poor, what will happen? 2) how many intermediate rewards are reasonable? 3) will n intermediate rewards be better than m if n>m?....*
> > > >
> > > > (1) We have an ablation study in Appendix C.6 which investigates the effect of intermediate rewards (with different quality) including:
> > > > - State-of-the-art novelty-based intrinsic motivation methods (RND [7], ICM [8], NovelD [9]) based on self-supervision that are helpful in hard exploration RL tasks. These methods can drive the agent to visit states of interest when they are trained well.
> > > > - Constant intermediate rewards, which can be considered with lower-quality compared to the other methods, as it provides a uniform incentive (with constant values) for the agent to visit different states (mimics the behavior of $\epsilon$-greedy exploration).
> > > >
> > > > As shown in Figure 15 in Appendix C.6, constant intrinsic rewards underperform novelty-based intrinsic motivation methods, since it does not distinguish familiar/novel states well. On the other hand, it leads to a smaller $L_1$ error than GFlowNets, since it still provides an exploration incentive for the agent to visit different states, which is possible to mitigate the issue of getting trapped in a few modes. Note that if the design of intrinsic intermediate rewards is not reasonable at all, then it will hurt the learning.
> > > >
> > > > (2 and 3) We provide intrinsic intermediate rewards to the agent at each step, whose magnitude is adaptively determined by its curiosity level about the states (smaller for more familiar states and larger for novel states generally). One can also determine the number of intermediate rewards and also decides which edges to be incorporated with intrinsic intermediate rewards for different tasks. However, it requires additional domain knowledge and hand design, and also additional hyperparameter tuning, which is difficult and not scalable.
> > > >
> > > > *Q4: Learning with more supervision may not be so positive or beneficial.*
> > > >
> > > > On the one hand, we have conducted extensive experiments in the paper to evaluate the effectiveness of GAFlowNets. The experimental results show that it significantly improves the learning of GFlowNets in hard exploration tasks, and prevents it from getting trapped in a few modes (please also find the detailed reason in the answer for Q1). On the other hand, curiosity-driven exploration bonus (learned in a self-supervised way) has shown to be effective empirically in related works [6].
> > > >
> > > > ---
> > > > We hope that the explanations and clarifications could relieve your concern. We are also happy to provide further clarification if you have any additional concerns.
> > > >
> > > > ---
> > > > *References*
> > > >
> > > > [1] Bengio, E., Jain, M., Korablyov, M., Precup, D., & Bengio, Y. (2021). Flow network based generative models for non-iterative diverse candidate generation. Advances in Neural Information Processing Systems, 34, 27381-27394.
> > > >
> > > > [2] ​​Malkin, N., Jain, M., Bengio, E., Sun, C., & Bengio, Y. (2022). Trajectory Balance: Improved Credit Assignment in GFlowNets. Advances in Neural Information Processing Systems, 35.
> > > >
> > > > [3] Madan, K., Rector-Brooks, J., Korablyov, M., Bengio, E., Jain, M., Nica, A., ... & Malkin, N. (2022). Learning GFlowNets from partial episodes for improved convergence and stability. arXiv preprint arXiv:2209.12782.
> > > >
> > > > [4] Ryan, R. M., & Deci, E. L. (2000). Intrinsic and extrinsic motivations: Classic definitions and new directions. Contemporary educational psychology, 25(1), 54-67.
> > > >
> > > > [5] Silvia, P. J. (2012). Curiosity and motivation. The Oxford handbook of human motivation, 157-166.
> > > >
> > > > [6] Burda, Y., Edwards, H., Pathak, D., Storkey, A., Darrell, T., & Efros, A. A. (2018). Large-Scale Study of Curiosity-Driven Learning. In International Conference on Learning Representations.
> > > >
> > > > [7] Burda, Y., Edwards, H., Storkey, A., & Klimov, O. (2018). Exploration by random network distillation. In International Conference on Learning Representations.
> > > >
> > > > [8] Pathak, D., Agrawal, P., Efros, A. A., & Darrell, T. (2017). Curiosity-driven exploration by self-supervised prediction. In International conference on machine learning (pp. 2778-2787). PMLR.
> > > >
> > > > [9] Zhang, T., Xu, H., Wang, X., Wu, Y., Keutzer, K., Gonzalez, J. E., & Tian, Y. (2021). Noveld: A simple yet effective exploration criterion. Advances in Neural Information Processing Systems, 34, 25217-25230.
> > > >
> > > > [10] Bengio, Y., Deleu, T., Hu, E. J., Lahlou, S., Tiwari, M., & Bengio, E. (2021). Gflownet foundations. arXiv preprint arXiv:2111.09266.

---

> > > > > ### Comment · Reviewer_kXmH · 2022-11-20
> > > > > **The response is solid.**
> > > > >
> > > > > Cool! Good response. Thanks for your positive feedback. I kept the promise to improve the score. See you in ICLR. Good luck.

---

> > > > > > ### Author Response · Authors · 2022-11-20
> > > > > > **Thank you!**
> > > > > >
> > > > > > Thank you for increasing the score.
> > > > > >
> > > > > > We would also like to thank the reviewer once again for the time and effort in reviewing our work and going through the rebuttal, which helped to improve our paper!

---

### Official Review · Reviewer_WVnH · 2022-10-25

**Confidence:** 3
**Correctness:** 3
**Technical Novelty And Significance:** 3
**Empirical Novelty And Significance:** 2
**Recommendation:** 8

**Clarity, Quality, Novelty And Reproducibility:**

The objective, the design of the  proposed framework, the algorithm are clearly described. Enough experiments are done to support claims with enough  evaluations. Conditions of experiments are described in details that is.helpful for its reproducibility.

**Strength And Weaknesses:**

The strength of this paper is a novel learning framework to incorporate intermediate rewards into GFlowNets. Incorporating intermediate rewards into GFlowNets has been already proposed. This paper formulate the problem by clarifying existing unsolved  problems and proposes a novel approach.
The algorithm is described in details. Enough experiments with further ablation study.
This is not weakness, but more evaluations of differences of learned structures between GFlowNets and GAFlowNets would support claims more clearly

**Summary Of The Paper:**

This paper introduces  Generative Augmented Flow Networks (GAFlowNets) that incorporate intermediate rewards into GFlowNets. Original GFlowNets  learn from the reward of the terminal state, and do not consider intermediate rewards. That is a problem in environments with sparse rewards. For this problem taking intermediate feedback signals into account helps to provide an exploration. (Bengio et al., 2021b) already proposed to incorporate intermediate rewards in a state- based manner. This approach  can result in slower convergence and large bias empirically, although it can explore more broadly. Incorporating the edge-based augmentation for efficiency, this paper proposes a joint way to take both edge-based and state-based augmented flows into account. Experiments on the GridWorld and molecule domains show the effectiveness of the proposed framework.


**Summary Of The Review:**

This paper formulate the problem by clarifying existing unsolved  problems of GFlowNets and proposes a novel approach. This paper proposes a joint way to take both edge-based and state-based augmented flows into account.The paper is enough described and organized to support claims with good experimental results.

---

> ### Author Response · Authors · 2022-11-11
> **Response to Reviewer WVnH**
>
> We thank the reviewer for the careful reading and valuable feedback of our work, and we greatly appreciate the comments! We have updated the paper to address the reviewer's comments, where the revisions are marked in blue.
>
> *Q1: More evaluations of differences of learned structures between GFlowNets and GAFlowNets would support claims more clearly.*
>
> Thanks for your suggestion. We include additional visualization of the learned structures for GFlowNets and GAFlowNets considering the forward policies $P_F$ in Appendix C.2 (and also the sample distribution from GFlowNets and GAFlowNets). It is shown that GFlowNet can get trapped in a few modes, while GAFlowNet is able to discover all the modes. Visualization of the top molecules discovered by GFlowNets and GAFlowNets in the molecule generation domain is demonstrated in Figure 18 in Appendix C.9, which shows that GAFlowNets can discover more diverse and high-quality solutions.
>
> We thank the reviewer for the time and effort in reviewing our work! We would very much appreciate it if the reviewer can check our responses and the updates in the paper and let us know whether they address your concerns. We are happy to provide further clarification if you have any additional concerns.

---

### Official Review · Reviewer_eLGw · 2022-10-25

**Confidence:** 3
**Correctness:** 4
**Technical Novelty And Significance:** 3
**Empirical Novelty And Significance:** 3
**Recommendation:** 8

**Clarity, Quality, Novelty And Reproducibility:**

The paper provides a joint form of dealing with state and edge based GFlowNet using RND base intrinsic rewards which is novel to the best of my knowledge. The paper is clear and justifies experimentally the experimental claims, the exact network choices could be further explained to improve reproducibility of results. Will the code be made publicly available?



**Strength And Weaknesses:**

Strengths: this paper is clear and well written, it provides a self contained and comprehensive literature review, it motivates the problem well and provides compelling results.
Weaknesses:
Can the authors provide citations/derivation for eq 4,5 and 6 in particular the telescoping derivation?
The experiments mention a coefficient parameter alpha that is never mentioned in the theoretical presentation of the model? Where is this used?
How important is the choice of intrinsic motivation function, is it enough to recover more structured forms of exploration? How far ahead can the flow information capture relevant future rewards?
What are the main limitations of the GAFlowNets in exploration, can the authors frame the proposed approach in terms of where would this kind of exploration shine vs where would it fail?


**Summary Of The Paper:**

This paper proposes to incorporate intrinsic reward augmentation in GFlowNets to improve exploration in sparse reward settings. It extends the intrinsic rewards to both state and edge intrinsic rewards improving on convergence and mode discovery in GflowNets in Gridworld experiments and more complex molecule generation tasks.


**Summary Of The Review:**

The paper tackles a problem of lack of intrinsic rewards in GFlowNets and provides a joint edge and state based flow intrinsic reward that captures well different modes and converges faster.

---

> ### Author Response · Authors · 2022-11-11
> **Response to Reviewer eLGw**
>
> We thank the reviewer for the thoughtful reviews and valuable feedback, and the comments are greatly appreciated! We have updated the paper to address the reviewer's comments, where the revisions are marked in blue.
>
> *Q1: Can the authors provide citations/derivation for eq 4-6 in particular the telescoping derivation?*
>
> We have incorporated derivations and citations for Equations (4)-(6), and please find the elaborations in Appendix A.
>
> *Q2: The experiments mention a coefficient parameter alpha that is never mentioned in the theoretical presentation of the model? Where is this used?*
>
> $\alpha$ is the intrinsic reward scaling coefficient (a constant) which controls the degree of exploration that we used in implementation (its corresponding ablation study can be found in Figure 6(b)). Specifically, $\rm{intrinsic \ reward}=\alpha || \phi(\textbf{s}) - \bar{\phi}_{\rm random}(\textbf{s}) ||^2_2$. Thanks for pointing this out, and we have revised the first paragraph on page 5 to avoid confusion.
>
> *Q3: How important is the choice of intrinsic motivation function, is it enough to recover more structured forms of exploration?*
>
> We studied the effect of different types of intrinsic motivation functions for exploration in Appendix C.6 including recent algorithms RND [1], ICM [2], and NovelD [3]. Results show that they are all effective in improving the learning of GFlowNets due to their improved exploration ability, where RND leads to the smallest empirical $L_1$ error. Therefore, GAFlowNet is not very sensitive to the intrinsic motivation mechanism, and our framework is general and can consider different types of intrinsic rewards. We also find that it scales well to the more complex and structured molecule generation task. Visualizations of the learned structures can be found in Appendix C.2 and C.9.
>
> [1] Burda, Y., Edwards, H., Storkey, A., & Klimov, O. (2018). Exploration by random network distillation. In International Conference on Learning Representations.
>
> [2] Pathak, D., Agrawal, P., Efros, A. A., & Darrell, T. (2017). Curiosity-driven exploration by self-supervised prediction. In International conference on machine learning (pp. 2778-2787). PMLR.
>
> [3] Zhang, T., Xu, H., Wang, X., Wu, Y., Keutzer, K., Gonzalez, J. E., & Tian, Y. (2021). Noveld: A simple yet effective exploration criterion. Advances in Neural Information Processing Systems, 34, 25217-25230.
>
> *Q4: How far ahead can the flow information capture relevant future rewards?*
>
> This is a great question. For simplicity, consider a special case where there is only one way to reach a particular state $s$, then the directed graph $G$ (described in Section 2) turns to a directed tree. In this case, the state flow for GFlowNets is equivalent to the sum of all downstream terminal rewards. For GAFlowNets, it is equivalent to the sum of all downstream intermediate and terminal rewards. In the general case (not necessarily a directed tree), the flow converges to a share of the sum of downstream terminal rewards, such that the sum of the flows through trajectories leading to a particular terminal state equals the total reward associated with that state (including those collected along the paths leading there).
>
> *Q5: What are the main limitations of the GAFlowNets in exploration, can the authors frame the proposed approach in terms of where would this kind of exploration shine vs where would it fail?*
>
> When formulating intermediate rewards as intrinsic motivation for exploration, GAFlowNets are effective to improve the learning of GFlowNets in tasks where there are some regions of the state space that are undesirable to explore. This includes a number of common scenarios like the challenging sparse reward tasks studied in Section 5.1.1, and hard difficulty tasks with dense rewards as investigated in Appendix C.5. For some easy tasks where exploration is not a critical challenge, then the performance gain of GAFlowNets over GFlowNets will be smaller.
>
> *Q6: The exact network choices could be further explained to improve reproducibility of results. Will the code be made publicly available?*
>
> Detailed experimental setup including network choices and hyperparameters can be found in Appendix C.1.2 (where all baselines are implemented based on [4] by following its setup), and we include an additional explanation in Appendix C.1.2. We are working on cleaning up the code and will definitely make it public.
>
> [4] Bengio, E., Jain, M., Korablyov, M., Precup, D., & Bengio, Y. (2021). Flow network based generative models for non-iterative diverse candidate generation. Advances in Neural Information Processing Systems, 34, 27381-27394.
>
>
> We thank the reviewer for the time and effort in reviewing our work! We would very much appreciate it if the reviewer can check our responses and the updates in the paper and let us know whether they address your concerns. We are happy to provide further clarification if you have any additional concerns.

---

> > ### Comment · Reviewer_eLGw · 2022-11-17
> > **what are the main limitations of the approach? how are they related/improve over RND limitations and GFlownets limitations?**
> >
> > I would like to thank the authors for the careful response. I believe most of my concerns have been addressed except the one pertaining to the limitations. For instance under which difficult exploration tasks is this approach failing, is this sensitive to noise in transitions, do the intrinsic motivation rewards augmentation suffer from the same issues as RND? how does it perform over stochastic traps? and how sensitive is to graph mis-specification? What are the limits of the proposed approach and its vulnerabilities to hard exploration tasks, for instance for hard exploration graphs for varying graph connectivity. Do the rewards augmentation help? This is a suggestion to enrich the understanding of the improvements proposed.

---

> > > ### Author Response · Authors · 2022-11-18
> > > **Response to Reviewer eLGw**
> > >
> > > We thank the reviewer for the insightful reviews and the reply!
> > >
> > > *Q1: For instance under which difficult exploration tasks is this approach failing, is this sensitive to noise in transitions, do the intrinsic motivation rewards augmentation suffer from the same issues as RND? How does it perform over stochastic traps?*
> > >
> > > Thanks for pointing this out. Our approach relies on the intrinsic motivation mechanism, here RND, and thus here will suffer from the limitations already existing with RND. However, if an improvement to RND or a better curiosity reward was designed, it could just be plugged-in to our methodology as GAFlowNet is a general framework. It is interesting for future work of GAFlowNets to investigate alternatives to RND that better handle aleatoric uncertainty.
> > >
> > > *Q2: How sensitive is to graph mis-specification? What are the limits of the proposed approach and its vulnerabilities to hard exploration tasks, for instance for hard exploration graphs for varying graph connectivity. Do the rewards augmentation help?*
> > >
> > > In the case where we generate molecular graphs or other kinds of graphs with a GFlowNet, the user of the algorithm is not specifying any graph [1]. It is the learner that generates graphs. So we are not sure what the reviewer mean by graph mis-specification. What could be designed by the user is the action space, that constrains the kinds of graphs that can be generated, but that is a design decision similar to designing the action space in RL in general (and it could be wrong for the same reasons). If you mean by graph the DAG (the set of all possible trajectories), it is indeed fully defined by the action space (which may restrict the set of allowed actions in particular states, e.g., an agent cannot go through a wall, or only two compatible atoms can be bound to form a molecule). In settings like the molecule generation (that we experimented with), the number of possible actions in each state indeed varies quite a lot. Indeed, our experiments demonstrate that our method achieves substantial performance improvement in these tasks.
> > >
> > > ---
> > > We thank the reviewer once again for the thoughtful reviews and fruitful discussion! We are also happy to provide further clarification if you have any additional concerns.
> > >
> > > ---
> > > *References*
> > >
> > > [1] Bengio, Y., Deleu, T., Hu, E. J., Lahlou, S., Tiwari, M., & Bengio, E. (2021). Gflownet foundations. arXiv preprint arXiv:2111.09266.

---

### Decision · Program_Chairs · 2023-01-20

**Decision:**

Accept: notable-top-25%

**Justification For Why Not Higher Score:**

The technical ideas are relevant to a specific community within the field, who will need to dive deep into the paper to understand the details.

**Justification For Why Not Lower Score:**

This is a neat paper that will benefit from a spotlight so that more people in the community will gain awareness of GFlowNets and related ideas.

**Metareview: Summary, Strengths And Weaknesses:**

This paper proposed Generative Augmented FlowNets, which are a variation of GFlowNets that can incorporate intermediate rewards to aid intrinsic exploration. Specifically, the paper combines edge-based augmentation with state-based augmentation proposed in prior work to improve convergence and mode discovery. The empirical results on grid world and molecule generation are very promising, and outperform several baselines while discovering a diverse set of solutions.
The reviewers were impressed with novelty of the solution and the strong results. There were some concerns around clarity of descriptions of intrinsic rewards and the augmentations but the author rebuttal seems to address those. I encourage the authors to make these revisions to strengthen their paper.

**Note From Pc:**

if the above contains the word "oral" or "spotlight" please see: "oral" presentation means -> notable-top-5% and "spotlight" means -> notable-top-25%. As stated in our emails, we are disassociating presentation type from AC recommendations